

# Systematizing and addressing theory uncertainties of unitarization with the Inverse Amplitude Method

**Alexandre Salas-Bernárdez[1][*], Felipe J. Llanes-Estrada[1],**
**Jose Antonio Oller[2] and Juan Escudero-Pedrosa[1]**

**1** Universidad Complutense de Madrid,
Depto. Física Teórica and IPARCOS, 28040 Madrid, Spain
**2** Universidad de Murcia, Departamento de Física, E-30071 Murcia, Spain

⋆ alexsala@ucm.es

## Abstract

**Effective Field Theories (EFTs) constructed as derivative expansions in powers of momentum, in the spirit of Chiral Perturbation Theory (ChPT), are a controllable approximation to strong dynamics as long as the energy of the interacting particles remains small, as they do not respect exact elastic unitarity. This limits their predictive power towards new physics at a higher scale if small separations from the Standard Model are found at the LHC or elsewhere. Unitarized chiral perturbation theory techniques have been devised to extend the reach of the EFT to regimes where partial waves are saturating unitarity, but their uncertainties have hitherto not been addressed thoroughly. Here we take one of the best known of them, the Inverse Amplitude Method (IAM), and carefully following its derivation, we quantify the uncertainty introduced at each step. We compare its hadron ChPT and its electroweak sector Higgs EFT applications. We find that the relative theoretical uncertainty of the IAM at the mass of the first resonance encountered in a partial-wave is of the same order in the counting as the starting uncertainty of the EFT at near-threshold energies, so that its unitarized extension should *a priori* be expected to be reasonably successful. This is so provided a check for zeroes of the partial wave amplitude is carried out and, if they appear near the resonance region, we show how to modify adequately the IAM to take them into account.**



# 1   Introduction

The LHC is preparing for its High-Luminosity Run (HL-LHC). It is possible that any new physics scale is beyond the reach of the collider and no new particles are found. Still, if there is a new scale not too much higher up in energy, its effects can be felt through the coefficients of an Effective Field Theory based on the Standard Model particles. These coefficients, typically multiplied by powers of the momentum that grow with the reaction energy, eventually entail unitarity violations as is well known from hadron physics. Chiral perturbation theory offers there a model-independent characterization of $\pi$, $K$ and $\eta$ interactions in the shape of an expansion on meson masses and momenta due to derivative couplings in the effective Lagrangian. The use of ChPT is restricted to energy ranges of about 200 MeV above the first production threshold. Perturbation theory based on that Effective Theory then fails, again due to fast unitarity violations. These are well known to appear in the resonant $J = 0,1$ $\pi\pi$ phase shifts [1, 2], with the low-energy scalar resonance $f_0(500)$ around 500 MeV (the threshold being nearby at 280 MeV); in $\eta \to 3\pi$ decays [3]; in $\gamma\gamma \to \pi^0\pi^0$ [4], etc.

Unitarization techniques such as the Inverse Amplitude Method (IAM) reviewed in this article allow computation of amplitudes at higher energies, at least up to and including the first resonance in each channel, (although often more resonances can also be generated like the $f_0(500)$ and $f_0(980)$ in the isoscalar scalar mesonic sector [5–7].) The same method has been deployed for Higgs Effective Field Theory (HEFT) in much recent work.

More generically, a widely used strategy in hadron physics [8] is to construct "unitarized" amplitudes that extrapolate to those higher energy regimes satisfying unitarity exactly (for elastic processes). Popular ones are the K-matrix method, that is already being incorporated into the Monte Carlo simulations of high-energy processes at the LHC; the $N/D$ method [9,10]; or variations of the Bethe-Salpeter equation [7,11]. The approach has not yet been widely adopted by experimental collaborations, but the lack of a yet higher energy collider means that the extrapolation of electroweak results by means of unitarity remains on the table. Among the problems to circumvent [12] is the fact that unitarity is best expressed in terms of partial waves, while the simulation chain of high-energy experiments is based on Feynman diagrams.

The most powerful unitarization methods are based on dispersion relations, incorporating known analyticity properties of scattering amplitudes. These methods solidly extrapolate the low-energy theory to the resonance region; a noteworthy approach among them, which has been broadly used, is the Inverse Amplitude Method [1] (IAM) [15–18].

In these dispersive approaches, shortly described in section 2, the Effective Theory is used to fix the subtraction constants of the dispersion relation, and because this relation can incorporate in principle all the model-independent information that first principles impose on the amplitude, the information contained in those low-energy coefficients is maximally exploited, generating an energy dependence valid at much higher values of the energy than originally expected.

For example, one can predict the mass, width, and couplings of the first resonance of the $W_L W_L$ scattering amplitude [19] (a tell-tale of what Higgs mechanism is at work, as the equivalence theorem guarantees that the Goldstone boson scattering amplitude coincides with that for $W_L W_L$ at high energy), for each angular momentum, $J$, and weak isospin, $I$, channel.

A check on the reliability of the IAM has been carried out in [20] by eliminating a heavy scalar particle from the theory and trying to reconstruct its mass (successfully) from its imprint in the low-energy parameters (see also [21]). To date however, the uncertainties of the dispersive approaches have not systematically been laid out, so that for some colleagues there remain question marks about the reliability of the unitarization methods. A classic way of analyzing different unitarization methods is to use several of them for the same problem and with the same perturbative amplitude, to map out a reasonable spread of possible results [22,23]. Instead, we try to follow the strategy of trying to *a priori* constrain the uncertainty in the unitarization method.

It is the purpose of this work to start analyzing the systematic uncertainties of the Inverse Amplitude Method to put its eventual predictions (if ever any separations from the SM couplings at the HL-LHC are discovered) on a firmer footing [2].

What we here undertake is to carefully review the derivation of the IAM method (section 3) and to discuss the uncertainty which may be assigned to each of the approximations therein (section 4). Section 5 presents a minimal outlook and distills the main conclusions of the analysis in Table 3. A few more details and derivations are left for an appendix.

---

[1]Truong has provided an interesting historic perspective of early developments [13]. For a recent review devoted to the unitarization techniques in general the reader can consult [14].

[2]The reader may wonder why not adopt the more precise Roy equations: the reason is that they require abundant data over the resonance region and beyond, which does not currently look like a reasonable expectation at any new electroweak physics sector at the LHC.

## 2 Unitarity and Dispersion Relations

### 2.1 Reminder of unitarity for partial waves

Conservation of wavefunction-probability in two-particle to two-particle scattering processes is expressed as a nonlinear integral relation for the scattering matrix $T_I(s, t, u)$ (where $s$, $t$ and $u$ are the well known Mandelstam variables and $I$ the isospin index) of definite isospin (in hadron physics) or electroweak isospin (in Higgs Effective Theory).

If we decompose $T_I$ in terms of partial waves of angular momentum $J$, the expression of unitarity is much simpler, see Eq. (3). This decomposition reads

$$T_I(s, t, u) = 16\eta\pi \sum_{J=0}^{\infty} (2J+1) t_{IJ}(s) P_J(\cos\theta_s) \tag{1}$$

and converges for physical $s$ and scattering angle $x \equiv \cos\theta_s \in [-1, 1]$; additionally, convergence succeeds over the Lehmann ellipse for unphysical $\cos\theta$ where the behavior of the Legendre polynomials $P_J$ is controllable [24]. If the scattering particles are identical, $\eta = 2$, otherwise $\eta = 1$. The explicit expression for the partial waves $t_{IJ}(s)$ is, inverting Eq. (1),

$$t_{IJ}(s) = \frac{1}{32\pi\eta} \int_{-1}^{+1} dx\, P_J(x) T_I(s, t(x), u(x)) . \tag{2}$$

The analytic structure of the amplitude in terms of the complex variable $s$ in the physical or first Riemann sheet is so that $T_I(s, t, u)$, as well as $t_{IJ}(s)$, are real in a segment of the real axis and they develop cuts, (and eventually bound state poles, though not in the physical systems here considered). For identical particles of mass $m_\pi$, such as in $\pi\pi$ scattering or $W_L W_L$ scattering, one such cut develops from the production threshold $s_{th} = 4m_\pi^2$, up to $+\infty$. (Sometimes the approximation $s_{th} \simeq 0$ is adopted.) This discontinuity is referred to as the right cut (RC). In the case that two pions appear also in the final state we will have that the partial waves inherit the same analytic structure as the amplitude, which, by crossing symmetry, develops a branch cut in the whole negative real axis which we call left cut (LC). The LC comes from the regions where physical particles are exchanged in the $t$- and $u$-channels and there is no possibility of avoiding the singularities of the amplitude (i.e. at the endpoints $x = -1, +1$). Inelastic channels, due to $2n$-pion states, $K\bar{K}$, $\eta\eta$, etc. in hadron physics, or $2n$-$\omega_i$, $hh$, etc. in the electroweak sector, would accrue additional cuts extending from $(2n)^2 m_\pi^2$, $4m_K^2$, $4m_\eta^2$, ... to $+\infty$, respectively.

The unitarity of the $S$ matrix makes the partial waves, for physical $Re(s) > 0$, obey the relation, akin to the optical theorem,

$$\text{Im}\, t_{IJ}(s) = \sigma(s)|t_{IJ}(s)|^2 , \tag{3}$$

which has the merit of being purely algebraic, though nonlinear in the partial waves. Here $\sigma(s) = \sqrt{1 - 4m^2/s}$ is the phase-space factor (which equals one for massless incoming particles). Since Eq. (3) is nonlinear, it restricts the modulus of the amplitude to satisfy

$$|t_{IJ}(s)| \le 1/\sigma(s) , \tag{4}$$

for physical $s$ above threshold.

The key observation for the Inverse Amplitude Method is that the unitarity condition for purely elastic processes in (3) also fixes the inverse of the partial wave for physical values of $s$ above the first threshold at $s_{th}$ as

$$\text{Im}\, \frac{1}{t_{IJ}(s)} = -\sigma(s) \text{ for } s > s_{th} . \tag{5}$$

This is a remarkable exact statement about a non-perturbative amplitude inasmuch as inelastic channels such as $W_L W_L \to W_L W_L W_L W_L$ for HEFT or $\pi\pi \to \pi\pi\pi\pi$ in hadron physics can be neglected and we will dedicate subsection 4.3 to assess the uncertainty due to this hypothesis, which is exact below the four-particle threshold and deteriorates as energy sufficiently increases beyond that. If the inelasticity stems from an additional two-body channel such as $K\bar{K}$ in hadron physics or $hh$ in the HEFT, the IAM requires a matrix extension. A brief recount is provided in subsection 4.2 below.

The immediate question that comes to mind, given that the imaginary part of the inverse amplitude is known from kinematics alone, is how can one bring in the dynamical information to also obtain its real part. To this end we dedicate the next subsection 2.2.

## 2.2 Exploiting the analytical properties of the inverse amplitude

For over a century, dispersion relations have been known to link the imaginary and real parts of "causal" functions satisfying Cauchy's theorem in appropriate complex-$E$ (here, $s$) plane regions. To exhibit and exploit the analytic structure of the inverse amplitude $1/t_{IJ}(s)$ we will deploy the appropriate dispersion relation (see *e.g.* [25] for a recent detailed introduction and the book [26] for a pedagogical account). The Cauchy theorem can be applied to any function $f(s)$ which is analytic in a complex-plane domain.

Application of the theorem is convenient for the following integral,

$$I(s) \equiv \frac{1}{2\pi i} \int_C dz \frac{f(z)}{(z-\epsilon)^n (z-s)} \,, \tag{6}$$

where $f(z)$ is taken to have two branch cuts extending from $4m^2$ to $+\infty$ and from $0$ to $-\infty$ (just as the partial wave amplitude $t(s)$), and the contour of integration $C$ is taken as depicted in Fig. 1. In Eq. (6), $s$ is above the RC (i.e. $s$ stands for $s + i\epsilon$ with $s \geq 4m_\pi^2$).

If the function $f$ is polynomially bounded as $|z| \to \infty$ (which, though not the case for partial wave amplitudes that can diverge exponentially for composite particles [27], is satisfied for their inverse amplitude in Eq. (9) below, that does fall as $e^{-az}$ with $a$ fixed) we are able to neglect the contribution to Eq. (6) coming from the two large semi-circumferences in Fig. 1.

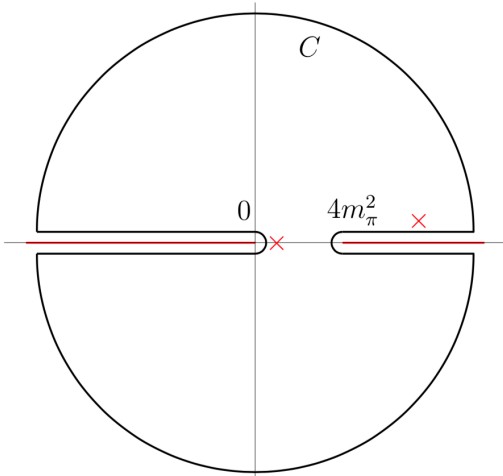

Figure 1: Analytic structure of elastic scattering partial waves for pions and the contour $C$ in the complex-$s$ plane that will be used to write a dispersion relation for the inverse amplitude. The red lines represent the discontinuity cuts in the partial wave amplitude. The red crosses additionally represent the $n$-th order pole at $z = \epsilon$ and the simple pole at $z = s + i\epsilon$ (with $s > 4m^2$) coming from the denominators in Eq. (6).

Due to Schwartz's reflection principle $f(s+i\epsilon) = f^*(s-i\epsilon)$, we are left in (6) with the integrals of the imaginary part of $f$ over the LC and RC,

$$I(s) = \frac{1}{\pi}\int_{-\infty}^{0} ds' \frac{\mathrm{Im}f(s')}{(s'-\epsilon)^n(s'-s)} + \frac{1}{\pi}\int_{4m_\pi^2}^{\infty} dz \frac{\mathrm{Im}f(s')}{(s'-\epsilon)^n(s'-s)} \,, \tag{7}$$

where $\mathrm{Im}f(s)$ is the imaginary part of $\lim_{\varepsilon\to 0^+} f(s+i\varepsilon)$ with $s \in$ RC or LC. On the other hand, $I(s)$ equals the sum of its residues coming from the simple pole at $z = s$ and the $n$-th order pole at $z = \epsilon$, with $\epsilon \in (0, 4m_\pi^2)$. In this way we find the $n$-times subtracted dispersion relation for $f(s)$,

$$f(s) = \sum_{k=0}^{n-1} \frac{f^{(k)}(\epsilon)}{k!}(s-\epsilon)^k + \frac{(s-\epsilon)^n}{\pi}\int_{-\infty}^{0} ds' \frac{\mathrm{Im}f(s')}{(s'-\epsilon)^n(s'-s)} + \frac{(s-\epsilon)^n}{\pi}\int_{4m_\pi^2}^{\infty} ds' \frac{\mathrm{Im}f(s')}{(s'-\epsilon)^n(s'-s)} \tag{8}$$

(which is valid safe at branch points where the multiple derivatives $f^{(k)}$ could fail to exist; this is generally of no concern).

## 3 The Inverse Amplitude Method: derivation

In ChPT the partial wave amplitude $t_{IJ}$ accepts a Taylor-like expansion in powers of $s$ (modified by logarithms) for small real $s$ as (dropping the $IJ$ subindices) $t \simeq t_0 + t_1 + \mathcal{O}(s^3)$, where $t_0 = a + bs$, and the leading behavior of each term in the series is $t_i \sim s^{i+1}$. Work in the seventies revealed the appeal of writing down a dispersion relation for the inverse amplitude [3] for pion-pion or electroweak Goldstone boson scattering, see [13,14,17,18,29]. It is customary since the last of these references to define the function, probably introduced by Lehmann [30],

$$G(s) \equiv \frac{t_0(s)^2}{t(s)} \,. \tag{9}$$

This function has the same analytic structure as $t_{IJ}$ except some additional poles coming from zeros of $t$. At low energies these zeros are known as Adler zeroes [31] and will indeed appear in scalar waves. For this function we make a third order subtraction (the order being the minimum compatible with the order of the EFT to which we work, given that the perturbative amplitudes' leading growth is polynomial) so that the dispersion relation for $G$ reads [4],

$$G(s) = G(\epsilon) + G'(\epsilon)(s-\epsilon) + \frac{1}{2}G''(\epsilon)(s-\epsilon)^2 + PC(G) +$$
$$+ \frac{(s-\epsilon)^3}{\pi}\int_{LC} ds' \frac{\mathrm{Im}\,G(s')}{(s'-\epsilon)^3(s'-s)} + \frac{(s-\epsilon)^3}{\pi}\int_{RC} ds' \frac{\mathrm{Im}\,G(s')}{(s'-\epsilon)^3(s'-s)} \,. \tag{10}$$

Proceeding backwards in this formula, we first encounter $PC(G)$, the contribution due to those Adler zeroes of $t$. The standard IAM method at Next to Leading Order (NLO) neglects their contribution at this order in ChPT since their size on the physical-$s$ half axis counts as NNLO [32]. However, there is no special difficulty in including them: the uncertainty introduced by neglecting the Adler zeroes is "*much less than the uncertainties (mostly of systematic origin) of*

---

[3] With equal right one could, instead of expanding $t \simeq t_0 + t_1$, expand $t^{-1} \simeq \frac{1}{t_0+t_1} \simeq \frac{1}{t_0} - \frac{t_1}{t_0^2}$ whose inverse leads directly to Eq. (12) below; but this expansion, while a direct mnemotecnic rule, is less conducive to an analysis of the uncertainties incurred, and does not expose the validity of the IAM in the complex $s$-plane as the dispersive derivation does, so we adopt the dispersive framework [28].

[4] *Strictu senso* we choose a subtraction point slightly separated from $s = 0$ so that the factors become $(s-\epsilon)$ and $\frac{1}{z-\epsilon}$ to avoid the divergence at $z = 0$ which is included in the interval of integration of the left cut. This plays little role in the derivation that follows.

*the existing data on meson-meson scattering*" [32] and that modified methods taking these into account differ in $\mathcal{O}(10^{-3})$ from the standard IAM in the physical region (see subsection 4.1 where we delve on the uncertainty in neglecting these zeroes and its remedy). Putting those Adler zeroes ("Pole Contributions" or $PC$) aside for now, we continue examining the remaining contributions to (10).

Second, the right cut (RC) of the integral is treated exactly by the IAM as long as only the elastic two-body cut contributes, and this is the one that the basic IAM includes. Inelasticities can be due to both two and four-body additional channels. The two-body ones can be treated with a coupled-channel IAM [33, 34], that has a less crisp theoretical basis: here we adopt the philosophy of staying within the one-channel IAM and use its coupled channel extension to estimate the uncertainty from omitting that channel, as long as this is sensible (see subsection 4.2). The four-body channel, as far as we know, is not tractable, so the IAM just leaves it out; but its unknown contribution can be controlled as it is short of phase space until quite high energies (see subsection 4.3).

Third, proceeding to the subtraction constants, we adopt three of them $G(\epsilon)$, $G'(\epsilon)$ and $G''(\epsilon)$ which suffices in hadron physics thanks to the saturation of the known Froissart-Martin bound [35, 36] controlling the growth of the physical cross section and the polynomial behavior of $t_0^2 \propto s^2$. In the NLO IAM, their values are taken from NLO ChPT, a valid approximation because they are taken with $s$ around zero, where the EFT is valid (the uncertainty therein is discussed in subsection 4.4). In this NLO approximation we can safely set $\epsilon = 0$ in the argument of the subtraction constants since $t_0$ and $t_1$ are essentially polynomials at such low $s$. This results in,

$$G(s) \equiv \frac{t_0(s)^2}{t(s)} = t_0(s) - t_1(s) + \frac{s^3}{\pi} \int_{LC} ds' \frac{\operatorname{Im} G(s') + \operatorname{Im} t_1(s')}{s'^3(s'-s)} . \tag{11}$$

In the fourth place and finally, this dispersion relation can be further simplified if we approximate the left cut contribution by taking the NLO chiral approximation of the discontinuity of $G$, $\operatorname{Im} G \simeq -\operatorname{Im} t_1$. Then the integral vanishes and a remarkable formula,

$$t_{IAM} \equiv \frac{t_0^2}{(t_0 - t_1)}, \tag{12}$$

is obtained: the usual IAM amplitude at NLO. In this step of the derivation, an uncertainty is introduced upon approximating the left cut, which is further examined in subsection 4.5 below. This is the most difficult one, as the left cut extends to $s = -\infty$ where the EFT is not valid: replacing $\operatorname{Im} G$ by $-\operatorname{Im} t_1$ is only sensible if the amplitude is wanted on the right-hand complex plane with $\operatorname{Im} s > 0$ where the influence of the left cut is smaller. It must be this larger distance between the left cut and the resonance region over right cut that binds the introduced uncertainty. This smaller contribution of the left-hand cut is readily observed when numerically comparing it to that of the right cut for the IAM amplitude itself at the $\rho$-resonance mass $m_\rho = 770$ MeV,

$$\int_{-\Lambda^2}^{-\lambda^2} dz \frac{\operatorname{Im} t_{IAM}(z)}{z^3(z - m_\rho^2)} \Big/ \left| \int_{4m_\pi^2}^{\Lambda^2} dz \frac{\operatorname{Im} t_{IAM}(z)}{z^3(z - m_\rho^2)} \right| \simeq 0.5\%, \tag{13}$$

where we choose $\lambda = 470$ MeV, i.e. the scale where ChPT is known to be reasonably accurate (see Fig. 5), and the cutoff as $\Lambda = 20$ GeV (where the value of the integrals becomes independent of this cutoff). This very small value of the uncertainty is however obtained by assuming the IAM also along the left cut, which is an unwarranted use thereof: we will later strive to obtain *a priori* bounds that do not assume the IAM's validity there.

The central quantity for the discussion in this article is the relative separation between the approximate IAM amplitude $t_{IAM}$ and the exact one, $t$, which $t_{IAM}$ approximates,

$$\Delta(s) = \left( \frac{t_{IAM}(s) - t(s)}{t_{IAM}(s)} \right). \tag{14}$$

Therein, the contribution due to approximating the LC follows from Eq. (11) to be

$$\Delta(s)G(s) \equiv \left( \frac{t_{IAM} - t}{t_{IAM}} \right) \frac{t_0^2}{t} = \frac{s^3}{\pi} \int_{LC} ds' \frac{\operatorname{Im} G + \operatorname{Im} t_1}{s'^3(s'-s)} . \tag{15}$$

Trying to set bounds on this integral is the goal of subsection 4.5.

When $t(s)$ is of slow variation, there is a chance that the relative uncertainty $\Delta(s)$ is numerically small. However, near a narrow resonance such as the $\rho$ or an equivalent $Z'$-like one in the electroweak sector, the amplitude is very sensitive to small changes of the pole position. There, $\Delta(m_\rho^2)$ can be of order 1, which is not very relevant: what is interesting then is to constrain the uncertainty incurred in computing that pole position, that is, the mass and width of the resonance.

We can discuss the position of a resonance in two ways. If it is isolated and narrow, a first approximation to $s_R$ is to use the saturation of unitarity $|t(s_R)| = 1/\sigma(s_R)$ over the real, physical $s$-axis. In the IAM, this reduces to solving for $s_R$ the simple algebraic equation [37]

$$t_0(s_R) - t_1(s_R) + i\sigma(s_R)t_0^2(s_R) = 0 , \tag{16}$$

which is equivalent to

$$t_0(s_R) - \operatorname{Re} t_1(s_R) = 0 . \tag{17}$$

The uncertainty introduced by this IAM approximation instead of employing the exact amplitude is equivalent to a nonvanishing quantity on the right hand side (RHS),

$$t_0(s_R) - t_1(s_R) + i\sigma(s_R)t_0^2(s_R) = -\Delta(s_R)G(s_R) . \tag{18}$$

Where all the functions are evaluated over the real $s$ RC.

We see therein that constraining the uncertainty of the amplitude ($\Delta$) also helps in constraining the uncertainty in the position of any resonance.

If one is willing to discuss $s_R$ with an imaginary part, an alternative starting point could be the relation between an amplitude in the first and in the second Riemann sheets $t^{II}(s) = \frac{-t^I(s)}{1+2i\sigma t^I(s)}$ so that the complex position of the pole can be obtained from the amplitude in the first sheet by $t(s_R) = \frac{i}{2\sigma}$. This yields a variant of Eq. (16)

$$t_0(s_R) - t_1(s_R) + 2i\sigma(s_R)t_0^2(s_R) = 0 , \tag{19}$$

but now for $s_R$ complex (in this case we have to choose the determination $\sigma(s^*) = -\sigma(s)^*$), with the equivalent of Eq. (18) being

$$t_0(s_R) - t_1(s_R) + 2i\sigma(s_R)t_0^2(s_R) = -\Delta(s_R)G(s_R) . \tag{20}$$

Either can be used to discuss new-physics resonances and, once one chosen, the relative uncertainties of one or the other method are very similar.

We will try to quantify the uncertainty in the position of a resonance following Eq. (18); and we will exemplify with the vector-isovector resonance in hadron physics (where all quantities are known) except in subsections 4.2 and 4.3 where the mass versus masslessness of the pions/Goldstone bosons in ChPT/HEFT respectively, do make a difference due to phase space.

# 4 Sources and estimates of uncertainty

As we have shown in section 3, the basic NLO-IAM treatment disregards contributions from Adler zeroes, from higher orders in perturbation theory, and from inelastic channels; and it approximates the contribution of the left cut. We wish to compute how the mass (and, only briefly around figure 12 below, the width) of a higher-energy resonance depends on these contributions and eventually put bounds on them to have systematic uncertainties under control. If we recover the full expression for $G(s)$ taking into account inelasticities, Adler zeroes and the next order in PT for the subtraction constants, we have to add terms to Eq. (15) to read

$$\Delta(s)G(s) = \frac{s^3}{\pi}\big(LC(G + t_1)\big) + 3^{rd}PT + PC(G) + \mathcal{I}_z\,, \tag{21}$$

where $\mathcal{I}_z$ takes into account the correction coming from inelasticities on the discontinuity of $t$ over the right cut (see the subsections 4.2 and 4.3 and specifically equation (33)), $LC$ represents the three-times-subtracted left-cut integral of the discontinuity in its argument, $3^{rd}PT$ takes into account the next order in PT correction to the displacement of the pole (see subsection 4.4)).

## 4.1 Uncertainty when neglecting the poles of the inverse $G = t_0^2/t$ (Adler zeroes of $t$ and CDD poles)

In deriving Eq. (12) through the inverse amplitude $1/t$, the possible zeroes of the amplitude in Eq. (10) were neglected. In chiral perturbation theory, such zeroes appear when one of the pions is taken with (near) zero mass and (near) zero energy and are referred to as "Adler zeroes" [31]. Beyond these, one could also have extra zeroes which we distinctively refer to as "Castillejo-Dyson-Dalitz" poles [38], and we treat both in the next two paragraphs. Such a CDD pole can be seen as a zero of $t$ not predicted by a subtracted dispersion relation, given both discontinuities along the LHC and RHC and fixing the subtraction constants by some known values of $T$. That is, dispersion relations do not have unique solutions and the CDD poles reflect it [38, 39]. In this sense, we consider that indeed an Adler zero is also a CDD pole, because it is LO in ChPT, while the discontinuities are at least NLO, unable to drive the correct energy dependence of the Adler zero.

### 4.1.1 Adler zeroes

This first shortcoming of the Adler zeroes was addressed in [32] and demonstrated to be quantitatively small; moreover, it is a systematic uncertainty that can be disposed of if extremely high precision was needed. In view of the larger ones that follow, we believe that this is unnecessary and we will limit ourselves to discussing it in this subsection, ignoring it thereafter.

Near threshold, the amplitude accepts the chiral expansion in Effective Theory, $t \simeq t_0 + t_1 + t_2 \cdots \simeq a + bs + \varepsilon s^2 + \dots$ up to logarithms. $1/t$ does not exist when $t$ vanishes; at LO, this happens at the Adler zero of $t$ that lies at $s = -a/b$. Keeping higher orders, its position slightly shifts. The inverse amplitude develops a pole there, but its effect is tiny once $s$ becomes even marginally larger than this value. The reason is that the dispersion relation is intelligently formulated in terms of $G = t_0^2/t$.

First, let us examine the chiral limit ($a = 0$) around which the effective theory is built. $G$ becomes a function of $s$ alone, independent of $m_\pi$ or, schematically, $G \simeq b^2 s^2/(s(b + \varepsilon s)) = b^2 s/(b + \varepsilon s)$ and the numerator's $s^2$ power has eliminated the zero of the denominator at threshold and there is no such pole.

Outside the chiral limit ($a \propto m_\pi^2$)

$$G \simeq \frac{(a+bs)^2}{a+bs+\varepsilon s^2} \, . \tag{22}$$

Once more, at LO, $G \simeq t_0^2/(t_0 + t_1) \simeq t_0$ presents no pole.

But the LO is insufficient near the zero of the denominator when $t_0$ and $t_1$ are nearly cancelling out. Then, the double zero of the numerator is slightly displaced with respect to the zero of the denominator, and the pole at low-$s$ is not exactly cancelled. To see it, let us factorize the denominator of $G$,

$$G \simeq b^2 \frac{(s+a/b)^2}{\varepsilon(s-s_+)(s-s_-)} \, , \tag{23}$$

with $s_\pm = -\frac{b}{2\varepsilon}(1 \pm \sqrt{1-4\varepsilon a/b^2})$. Taking into account the chiral counting, $a \sim m_\pi^2$, $\varepsilon \sim 1/\Lambda^2$, we see that there is a pole near the original one, $s_- \simeq -\frac{a}{b} + O\left(\frac{m_\pi^4}{\Lambda^2}\right)$ and a pole that comes from infinity, at $s_+ \simeq -\frac{b}{\varepsilon} \sim O(\Lambda^2)$. This last one is outside the range of validity of the theory and can be safely discarded. The first one is unavoidable, but remains below the threshold, and when $s$ is in the physical zone, its effect on $G$ is of order $\frac{m_\pi^4}{s\Lambda^2}$: the memory of the Adler zero is very small outside its very proximity. Since the dispersion relation for the right cut is weighted for $s$ far from this zero of $t$ (pole of $G$), its presence becomes a numerically small correction.

Summarizing so far: the Adler zero causes no encumbrance in the chiral limit, which is often the approximation used for HEFT [22]; and if masses are kept, it introduces a very small uncertainty.

In any case, this one uncertainty of the basic method can be disposed of, if need be, by use of the "Modified Inverse Amplitude Method" of [32]. Here, the amplitude is represented by

$$t_{\mathrm{mIAM}} \equiv \frac{t_0^2}{t_0 - t_1 + A_{\mathrm{mIAM}}} \, , \tag{24}$$

with a modification term in the denominator that uses the position of the Adler zero computed in chiral perturbation theory (appropriate since $s$ is very low), $s_A \simeq s_0 + s_1 + \dots$

$$A_{\mathrm{mIAM}} = t_1(s_0) - \frac{(s_0 - s_A)(s-s_0)}{s-s_A}(t_0'(s_0) - t_1'(s_0)) \, . \tag{25}$$

At the position of a resonance, $s = s_R \gg s_0, s_A$, the Mandelstam variable $s$ drops out and $A_{\mathrm{mIAM}} \sim \mathcal{O}(s_0^2, s_1)$ produces a small constant shift of the pole position upon substituting it in the denominator of Eq. (25). The relative uncertainty in that pole position is therefore $\mathcal{O}(s_0^2/s_R^2)$.

The largest such uncertainty will affect the channel with the lightest resonance, the $IJ = 00$ scalar, isoscalar partial wave that has its Adler zero at $s_0 = m_\pi^2/2$ in ChPT (see for example Eq. (9.3.21) of [40] for the value of $s_1$). Thus, the uncertainty at the $f_0(500)$ pole in this scalar channel is of $\mathcal{O}(m_\pi^4/M_{f_0}^4) \simeq 0.6\%$, at the few per mille level in hadron physics. For the Electroweak Chiral Lagrangian or HEFT, taking $(m_Z/1\mathrm{TeV})^4 \simeq 7 \times 10^{-5}$, we see that the uncertainty is totally negligible.

### 4.1.2 Castillejo-Dalitz-Dyson poles of the inverse amplitude

It remains to discuss what happens if $t_0 + t_1$, now a polynomial of second order (up to a logarithm) develops an additional zero above the threshold. These zeroes give rise to so-called Castillejo-Dyson-Dalitz (CDD) poles [38] of the inverse amplitude. A dispersion relation for

$G \propto 1/t$ would need an additional pole contribution with treatment parallel to that of the Adler zero just discussed.

There is little that one can do to avoid these CDD poles from contributing if/when they are present, as they represent new physics which is neither obvious nor generated from the boson-boson scattering dynamics itself [10, 39]; nevertheless, we do not consider them a source of uncertainty since if they are separately treated: given the measurement of the low–energy coefficients, one can try to identify the presence of the CDD pole and subtract it as shown shortly.

Such CDD pole often appears in the $I = J = 0$ $\pi\pi$ partial wave: in many parametrizations, the phase shift is larger than $\pi$ below the $K\bar{K}$ threshold [41], where the amplitude is considered to be elastic. Therefore, both real and imaginary parts of $t$ vanish, $t(s) = \sigma e^{i\delta} \sin \delta$ is zero at the point $\delta(s_C) = \pi$ with $\sqrt{s_C} < 2m_K$.

The basic IAM runs into trouble when the zero of $t$ at the CDD pole happens near a resonance, because then two contradictory equations need to be satisfied: the resonance condition (vanishing of the denominator in $t_{IAM}(s)$ at $s = s_R$) $t_0(s_R) - \mathrm{Re} t_1(s_R) = 0$ and the CDD-pole condition from the ChPT expansion at $s = s_C$, $t_0(s_C) + \mathrm{Re} t_1(s_C) = 0$. Their simultaneous fulfillment would imply that $\mathrm{Re} t_1$ and $t_0$ vanish at close values of $s$, which is not possible anywhere near the resonance region, since $t_0(s)$ is zero only at the Adler zero which is below threshold. Not taking care of this CDD pole was found to make the prediction of a new resonance to be off by as much as 25% in Ref. [42], so care should be exercised. This is shown with an explicit example in Appendix A.2.

### 4.1.3 How to handle a CDD pole if present

The difficulty can be overcome by studying the behavior of $t_0 + t_1$ in each case and, if suspicion of a CDD pole of $G \propto t^{-1}$ arises, by appropriately modifying the IAM as we next sketch.

Hence, the first thing to do is to detect the possibility of a CDD pole from the low energy chiral expansion. This expansion is such that the imaginary part of $t_1$ does not vanish above $4m_\pi^2$, the two-pion threshold; thus, it is necessary to study its real part.

The question to be examined upon deploying the IAM for any partial wave is whether there is a value $s = s_C$ above threshold such that its real part does vanish

$$t_0(s_C) + \mathrm{Re} t_1(s_C) = 0 \, . \tag{26}$$

This is a practical test to try to detect the presence of a CDD pole. When focused on the toy-model-amplitude $t(s)$ of Eq. (82) in Appendix A.2, it actually yields the exact result, detecting the CDD pole of the inverse amplitude at $s_C = M_0^2$, as evident in Eq. (85). [5]

Of course, the presence of a zero in $t(s)$ requires both real and imaginary parts to vanish whereas Eq. (26) is a condition on the real part alone. One can ask to what extent it is a good criterion for detecting a zero in the partial wave from the knowledge of the NLO amplitude alone. The reason it is useful is that elastic partial waves $t(s) = e^{i\delta} \sin \delta$ only have one independent real function, the phase shift $\delta(s)$. $\mathrm{Re}(t(s)) = \cos \delta \sin \delta$ vanishes for $\delta = 0$ or $\pi/2$ mod $\pi$, but the second case corresponds to maximum (nonzero) imaginary part and a resonant amplitude, and nothing should be said about it from the purely perturbative $t$. Thus, $\delta = 0 \bmod \pi$, implying $\sin \delta = 0$ and $|t(s)| = 0$ is the only relevant case, and because the amplitude is small near the zero, Eq. (26) is generally adequate.

Once such CDD pole at $s_C$ has been identified, a second step is to introduce an auxiliary

---

[5]This example also illustrates that different results can be obtained upon looking for poles in the second Riemann sheet versus establishing the position of a resonance by requiring the vanishing of the real part of the partial wave.

function $t^C(s)$ without the related zero at $s_C$, that can be minimally defined as

$$t^C(s) = \frac{t(s)}{s - s_C} . \tag{27}$$

It is the real part of the inverse of this auxiliary function $\mathrm{Re}(1/t^C(s))$ that is chirally expanded up to $\mathcal{O}(s^2)$ or NLO. To do it, we note that $s_C = \mathcal{O}(p^0)$ because $s_C$ is not an Adler zero, it is a large scale. Such expansion gives

$$\mathrm{Re}\frac{1}{t^C(s)} = \mathrm{Re}\frac{s - s_C}{t_0 + t_1} = \frac{s - s_C}{t_0} + s_C\frac{\mathrm{Re}\,t_1}{t_0^2} + \mathcal{O}(s) . \tag{28}$$

In turn, the imaginary part of $1/t^C(s)$ is fixed to $-i(s - s_C)\sigma(s)$ by elastic two-body unitarity and, added to the real part, yields Eq. (29) below.

This discussion can be wrapped up in two simple routine steps: check for the presence of a CDD pole with Eq. (26), and if one is present, then make a substitution in the IAM,

$$t_{\mathrm{IAM}} = \frac{t_0^2}{t_0 - t_1} \rightarrow \frac{t_0^2}{t_0 - t_1 + \frac{s}{s - s_c}\mathrm{Re}(t_1)} . \tag{29}$$

The additional piece in the denominator of this equation, from applying the IAM to $t^C$ instead of $t$, guarantees the CDD pole at the correct position, and does not affect the good unitarity behavior that the IAM enjoys; in the chiral counting, the difference between both formulae starts at order $s^3$.

Applying this procedure to the toy amplitude of Eq. (82) gives (the next equality implying an approximation in chiral perturbation theory)

$$G^C := \frac{1}{t^C(s)} = \frac{\mathfrak{f}^4}{s} - i\sigma(s - s_C) , \tag{30}$$

from which one can immediately recover the exact $t(s) = (s - s_C)t^C(s)$.

Once the expansion of $t^C(s)$ is at hand, the IAM is applied thereto, and upon completion, the zero that was factored out is multiplied back to reconstruct the approximation to the amplitude. In the example in Appendix A.2 this is

$$t_{IAM}(s) = \frac{1}{\frac{\mathfrak{f}^4}{s(s - M_0^2)} - i\sigma} , \tag{31}$$

that indeed reproduces the exact $t(s)$ of Eq. (82). The conclusion, therefore, is that the unhandled presence of a CDD pole of the inverse amplitude leads to uncertainties of order $M_R^2/M_0^2 \sim \mathcal{O}(1)$, but these can be dealt with, analogously to the Adler zeroes, to eliminate the problem. This observation is elevated to table 3. Another example is also worked out in the Appendix A.2 for the $I = J = 1$ channel with HEFT [22], by choosing the low-energy constants to generate a zero in the partial-wave amplitude.

## 4.2 From additional two-body channels

Strictly speaking, the unitarity condition in Eq. (3) is valid below any inelastic threshold. Generically, in the presence of several elastic and inelastic channels, that unitarity relation should be modified as

$$\mathrm{Im}\,t = \sum_i \sigma_i(s)|t_{\pi\pi \to i}|^2 \theta(s - (\sum_j m_j)^2), \tag{32}$$

where $\sum_j m_j$ represents the sum of the masses of the intermediate state particles and $\sigma_i(s)$ the phase space factor, all in the $i$th channel (also, it is intended that the symbol $\sum_i$ sums or integrates over remaining quantum numbers in channel $i$).

The first (strongly interacting) channel to open, as allowed by $G$-parity, appears when four pions can go on-shell in the intermediate state (around 550 MeV). We discuss this in the next subsection 4.3.

Here we start by discussing the uncertainty introduced by the first *two-body* inelastic channel, with two kaons in the intermediate state, $K\bar{K}$, and with threshold around 985 MeV. Above this energy, the unitarity relation for the inverse amplitude in Eq. (5) is modified to read

$$\text{Im} \frac{1}{t_{\pi\pi}} = -\sigma_{\pi\pi}\Big(1 + \frac{\sigma_{K\bar{K}}}{\sigma_{\pi\pi}} \frac{|t_{\pi\pi\to K\bar{K}}|^2}{|t_{\pi\pi\to\pi\pi}|^2}\Big). \tag{33}$$

There are two regimes that allow to disregard the inelastic channel in this modified unitarity equation: when the ratio of the squared amplitudes is small, or when the ratio of the phase-space factors is the one suppressing the amplitude.

The second term inside the parenthesis of Eq. (33) is the correction to the right cut discontinuity of $t$ related to the inelastic "$\mathcal{I}_z$" contribution to Eq. (21),

$$\mathcal{I}_z(s) = \frac{s^3}{\pi} \int_{RC} dz \frac{-t_0^2 \sigma_{\pi\pi}}{z^3(z-s)} \frac{\sigma_{K\bar{K}}}{\sigma_{\pi\pi}} \frac{|t_{\pi\pi\to K\bar{K}}|^2}{|t_{\pi\pi\to\pi\pi}|^2}. \tag{34}$$

In this form we additionally see that the three subtractions bias the integral in Eq. (34) towards the low energy region so that the effect of the coupled channels is even less prominent if the elastic and inelastic amplitudes have the same scaling with $z$ so that its powers cancel out in the ratio leaving only logarithms (which is quite the case in ChPT). Finally, the $z-s$ factor in the denominator enhances the region of $z$ around the external value of $s$ which is the argument of $\mathcal{I}_z(s)$. (This we often take as the $\rho$-resonance mass, *i.e.* $m_\rho = 770$ MeV).

**In hadron physics,** all these mechanisms are suppressing the inelastic coupling of $\pi\pi$ and $K\bar{K}$ below 1.2 GeV [34], except near the $f_0(980)$ resonance [6]. Because those happen close to the $K\bar{K}$ threshold, and the first resonances in $\pi\pi$ scattering ($\sigma$ and $\rho$) lie below it, the uncertainty in the latter's masses is well controlled.

In practice, the dominant contribution to the integrated uncertainty $\mathcal{I}_z(s)$ of Eq. (34) in meson scattering comes from the energy region up to 1.2 GeV where we can constrain it with the available data for $|t_{\pi\pi\to K\bar{K}}|^2/|t_{\pi\pi\to\pi\pi}|^2$ in the $J = I = 1$ channel as the integrand will be heavily suppressed above (and much below) $m_\rho$. The factor

$$\frac{\sigma_{K\bar{K}}}{\sigma_{\pi\pi}} \frac{|t_{\pi\pi\to K\bar{K}}^{IJ=11}|^2}{|t_{\pi\pi\to\pi\pi}^{IJ=11}|^2} = \frac{\sigma_{\pi\pi}}{\sigma_{K\bar{K}}} \frac{1-\eta_{11}^2}{\eta_{11}^2 - 2\cos 2\delta_{\pi\pi} + 1} \tag{35}$$

is less than 0.08 below 1.2 GeV (because the vector-isovector elasticity, $\eta_{11}$, is relatively close to one in the 1-1.2 GeV energy region, $\eta_{11} \simeq 0.99$, [34, 43] and the pion-pion channel phase shift, $\delta_{\pi\pi}$, varies slowly in this energy region).

We will therefore concentrate on the one-channel IAM that entirely neglects the contribution of the second channel, and use the coupled-channel IAM only to estimate the uncertainty therein.

Introducing the value above allows us to set a bound to the displacement of the pole from the $K\bar{K}$ intermediate state up to $s = (1.2 \text{ GeV})^2$. Since we know that, below 1.2 GeV,

$$\frac{\sigma_{KK}|t_{\pi\pi\to K\bar{K}}^{IJ=11}|^2}{\sigma_{\pi\pi}|t_{\pi\pi\to\pi\pi}^{IJ=11}|^2} \le 0.08, \tag{36}$$

we can use Eq. (18), accounting for the uncertainty due to the two-body inelastic contribution to the displacement of the pole, $\mathcal{I}_z$ in (21), (34) and (36) above to find

$$|\Delta(s)G(s)| \leq 0.08\frac{s^3}{\pi}|RC(t_1)|(s) \simeq 1 \cdot 10^{-4} , \tag{37}$$

with the expression evaluated at $s = m_\rho^2$. (Though this is below the two-kaon threshold, so that at the resonance mass $\sigma_{KK} = 0$, the uncertainty does not vanish because the integral in $RC$ extends through $\infty$.) In Eq. (37) $RC(f)(s) = \int_{RC} dz\, \mathrm{Im}f(z)/[z^3(z-s)]$. We take such potential displacement of the $\rho$-pole due to this uncertainty to Table 3.

**Within the Electroweak Standard Model,** the coupled-channel IAM has also been deployed in a series of articles [44]. Because in the $s \sim 1\ \mathrm{TeV}^2$ region (where the HEFT would be used) the $W$, $Z$ and $h$ boson squared masses of order $0.01\ \mathrm{TeV}^2$ are all negligible, they have equal phase space (the ratio $\sigma_{hh}/\sigma_{\omega\omega}$ is close to 1), providing no suppression to Eq. (33); only the equivalent ratio of squared amplitudes $|t_{ww \to hh}|^2/|t_{ww \to ww}|^2$ suppresses the inter-channel coupling. This is, in the notation of [22], proportional to the parameter combination $(a^2-b)^2$, that vanishes in the Standard Model (but its value in strongly interacting theories of BSM physics is of course unknown). What we can state is that, if low-energy measurements reveal this combination of low-energy constants to be numerically small, the coupled-channel uncertainty introduced into the single-channel problem is immediately under control.

We lean on the IAM method for coupled channels to generate the uncertainty of the one-channel IAM. The coupled-channel case [6,34,45] takes the same form as the standard IAM but in terms of matrix-valued amplitudes. For $n$ coupled channels, the amplitude will be an $n$-by-$n$ matrix, $\mathbf{t}$, so that the IAM expression becomes

$$\mathbf{t}_{IAM} = \mathbf{t}_0(\mathbf{t}_0 - \mathbf{t}_1)^{-1}\mathbf{t}_0 . \tag{38}$$

Concentrating on the HEFT with two channels, upon disregard of the contributions to the IAM partial wave amplitude of $ww \to ww$ coming from the coupled channels $ww \to hh$ and $hh \to hh$, an error $\delta$ is incurred,

$$\delta = \frac{(t_1^{11}t_0^{12} - t_0^{11}t_1^{12})(t_1^{11}t_0^{21} - t_0^{11}t_1^{21})(t_0^{11} - t_1^{11})^{-1}}{(t_1^{12}(t_0^{21} - t_1^{21}) + t_0^{12}(t_1^{21} - t_0^{21}) + (t_0^{11} - t_1^{11})(t_0^{22} - t_1^{22}))} , \tag{39}$$

where the superindices denote matrix elements of $\mathbf{t}$. We are parametrizing $t_{IAM}^{11} = t_{IAM} + \delta$ so that $\delta$ takes into account all contributions from coupled channels to the one-channel IAM partial wave amplitude, $t_{IAM}$. Note that, neglecting logarithms, Eq. (39) is a fraction of second-order polynomials in $s$ and the zeroes of the denominator correspond to zeroes of $\det(\mathbf{t}_{IAM}(s)^{-1})$ and, therefore, to poles in the amplitudes (resonances).

Reading the coefficients from [22] for the $IJ = 00$ channel (neglecting logarithmic contributions), the expression for $\delta$ at LO in $(a^2-b)^2$ and $(3d+e)$ is

$$\delta(s) \simeq s^2(a^2-b)^2\left[\frac{-9(29(a^2-1)^2 + 5376\pi^2 a_4 + 8448\pi^2 a_5)^2}{2949440\pi g(576\pi^2 v^2(a^2-1) + 5376\pi^2 a_4 s + 8448\pi^2 a_5 s + 101(a^2-1)^2 s)^2}\right] . \tag{40}$$

We see from Eq. (40) that, as the coupling between channels $(a^2-b)$ approaches zero, the uncertainty from neglecting the coupled channel vanishes as expected. The denominator

Table 1: Bounds on the parameters of the HEFT Lagrangian and the allowed example values that we have used to estimate the uncertainty in Eq. (39). We write down a sensible value from considering the 95% confidence bounds, with no attempt at combining different experiments or theoretical analysis. Additionally, $v = 246$ GeV as is well known from the electroweak theory.

| HEFT Parameter | Bounds known to us | Example value taken |
|:---:|:---:|:---:|
| $|a-1|$ | $< 0.15$ [37, 46] | $a = 0.9$ |
| $b-1$ | $\in (-1, 3)$ [47, 48] | $b = 1.5$ |
| $a_4$ | $< 6 \cdot 10^{-4}$ [49] | $5 \cdot 10^{-4}$ |
| $a_5$ | $< 8 \cdot 10^{-4}$ [49] | $5 \cdot 10^{-4}$ |

of Eq. (40) carries a dependence on the *hhhh* interaction parameter $g$ coming from inverting the $G^{ij}$ matrix [6].

A particularity of this HEFT theory is that a small numerical factor suppresses $\delta$. Evaluating it for $s = 1$ TeV, with $g = 1$, and the example values in table 1 for the remaining coefficients of the HEFT Effective Lagrangian (consistent with current LHC bounds), we obtain a tiny

$$|\delta(s = 1 \text{ TeV}^2)| \simeq 5 \cdot 10^{-3} . \tag{41}$$

This two-body uncertainty propagates to the pole position of the $IJ = 00$ channel resonance, whose real part (new physics mass) becomes fuzzed by

$$|\Delta(s)G(s)| = \left| \frac{-\delta}{t_{IAM}} \frac{t_0^2}{t_{IAM} + \delta} \right| \simeq 2 \cdot 10^{-3} \tag{42}$$

at 1 TeV; this is an order of magnitude larger than the hadron physics counterpart in Eq. (37). It is not unexpected because of the heavy phase space suppression in the earlier case. Even so, with present knowledge, if a scale of new physics strongly affecting vector-boson scattering is discovered, the corresponding inelasticity does not appear to be a worry.

### 4.3 From inelastic channels with additional identical particles

Let us start discussing the case of hadron physics. As mentioned at the beginning of Section 4.2, a four-pion channel opens around 550 MeV. *A posteriori*, the accuracy of the IAM would be suggestive of a small contribution by the four-pion channel (a four-pion intermediate state is a three-loop and higher-order effect in ChPT), just as the $K\bar{K}$-one discussed in subsection 4.2. Phenomenologically, the effect of that $4\pi$ channel for $\sqrt{s} < 1$ GeV seems to be very small as indicated by a small experimental inelasticity $1 - \eta \approx 0$, as well as from explicit calculations [50]. We will address the uncertainty introduced by neglecting this channel *a priori*, basically controlling it due to reduced phase space.

The massive $n$-particle phase space differential is

$$d\phi_n = \delta^{(4)} \left( p - \sum_{i=1}^{n} q_i \right) \prod_{i=1}^{n} \frac{d^3 q_i}{(2\pi)^3 2E_i} . \tag{43}$$

Integrating this four particle phase space, having used the three-dimensional delta function, we arrive at

$$\phi_4 = \int_0^{+\infty} \left( \prod_{i=1}^{3} \frac{d|p_i||p_i|^2}{2E_i} \right) \int_{-1}^{+1} dx_1 dx_2 \int_0^{2\pi} d\varphi \times \frac{\delta(\sqrt{s} - E_1 - E_2 - E_3 - E_4)}{(2\pi)^6 E_4} . \tag{44}$$

---

[6]This cannot be set to zero for invertibility, but as the channels decouple its value becomes irrelevant; thus, we take this parameter $g$ to be of order one with little loss of precision on the $\omega\omega$ channel.

$$\text{Im } t \propto \phi_2 |t|^2 + \phi_4 |t_{2\to4}|^2 \propto$$

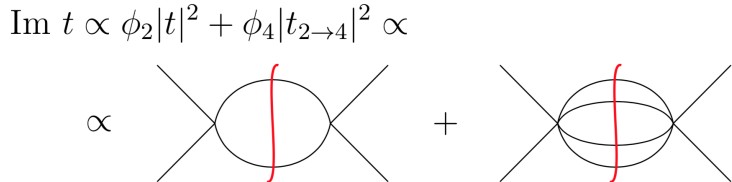

Figure 2: When considering each of the 2- or 4-body phase spaces, we have assigned $2\pi$ factors according to the normalization of one-pion states. For each pion in an intermediate state there is a factor $(2\pi)^{-4}$. However, when calculating the imaginary part of the amplitude, each cut line will produce an extra $2\pi$ (together with a $f_\pi^{-1}$). This leaves the typical $(2\pi)^{-3}$ normalization factor for each pion. Hence, we must only use $f_\pi^4$ to compare the two-body and four-body phase spaces.

Here

$$E_4 = \Big( |p_1|^2 + |p_2|^2 + |p_3|^2 + 2|p_1||p_2|x_2 + 2|p_2||p_3|x_3 +$$
$$+ 2|p_2||p_3|\Big( x_2 x_3 + sin\varphi \sqrt{1-x_2^2}\sqrt{1-x_2^2} \Big) + m^2 \Big)^{\frac{1}{2}}. \tag{45}$$

After numerically integrating the expression in Eq. (44), we can compare the four-particle phase phase, $\phi_4$, with the two-body one, $\phi_2$, which is related to $\sigma(s)$ defined in Eq. (3) as $\phi_2(s) = \sigma(s)/8\pi$. For a fair comparison, the two-body phase space needs to be multiplied by a power of $f_\pi$, $\phi_2 \times f_\pi^4$, to match the dimensions of the four-body one (here the pion decay constant is taken to be $f_\pi = 93$ MeV). This $f_\pi^4$ dimensionful factor is extracted from the two-body amplitude (that correspondingly has different dimensions from the inelastic two to four body one). The reasoning behind this choice is explained in Fig. 2.

The resulting ratio, which is the meaningful measure of the relative weigt of four- and two-body states, $\phi_4/(\phi_2 f_\pi^4)$, is plotted in Fig. 3. We see that, up to $s = (1.1 \text{ GeV})^2$, the four-particle phase space is heavily suppressed compared to the two particle one [7].

For the HEFT case in the TeV region it is an excellent approximation to adopt massless particles, which does away with the need for numerical computation since the $n$-particle phase space has a simple analytic expression

$$\phi_n = \frac{1}{2(4\pi)^{2n-3}} \frac{s^{n-2}}{\Gamma(n)\Gamma(n-1)} . \tag{46}$$

We plot in Fig. 4 the ratio analogous to figure 3, substituting $f$ by $v = 0.246$ TeV as appropriate for the electroweak sector. Therein we see that, up to 3 TeV, four particle phase space is again much smaller than the two particle phase space.

Now we can compute the displacement of the pole in both hadron physics and the electroweak HEFT, taking the ratio $|t_{\pi\pi\to\pi\pi\pi\pi}|^2/|t_{\pi\pi\to\pi\pi}|^2$ and $|t_{ww\to wwww}|^2/|t_{ww\to ww}|^2$ to be of order one, with the phase space ratios from Fig. 3 and Fig. 4. Taking this information to Eq. (21) and in analogy with Eq. (33) for the four body intermediate states, we find

$$|\Delta(s)G(s)| \le \begin{cases} 2 \cdot 10^{-2} \frac{s^3}{\pi} |RC(t_1)|(s) & \text{for hadron physics} \\ 8 \cdot 10^{-2} \frac{s^3}{\pi} |RC(t_1)|(s) & \text{for the HEFT .} \end{cases}$$

This means that for hadron physics, after computing the right cut integral, the channel $IJ = 11$ receives an uncertainty of order

$$|\Delta(s)G(s)| \le 4 \cdot 10^{-5} \quad \text{for } s = m_\rho^2 . \tag{47}$$

---

[7]This is easily understood by the analogous simple relation $\int_0^\infty dx_1 dx_2 dx_3 dx_4 \delta(\sum_i x_i - 1) < \int_0^\infty dy_1 dy_2 \delta(\sum_i y_i - 1)$.

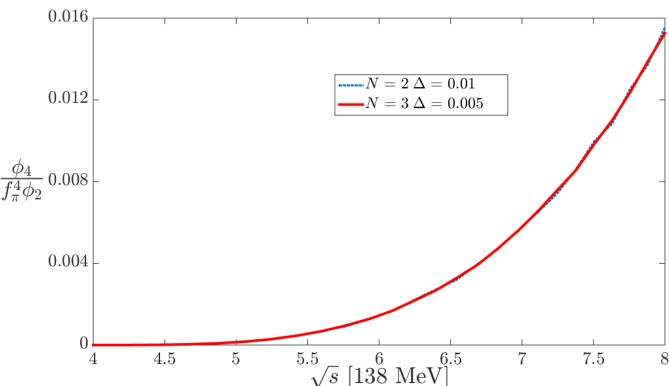

Figure 3: Numerical computation of the four- to two-particle phase-space ratio from threshold $\sqrt{s} = 4m_\pi$ up to $\sqrt{s} = 1.1$ GeV. (The 4-body integral is calculated in successively better approximation by replacing the energy delta function in Eq (44) by a Gaussian of decreasing width $\Delta$. $N$ is proportional to the number of 20-Gaussian point partitions of each variable's integration interval. Convergence is excellent and both curves are barely distinguishable.)

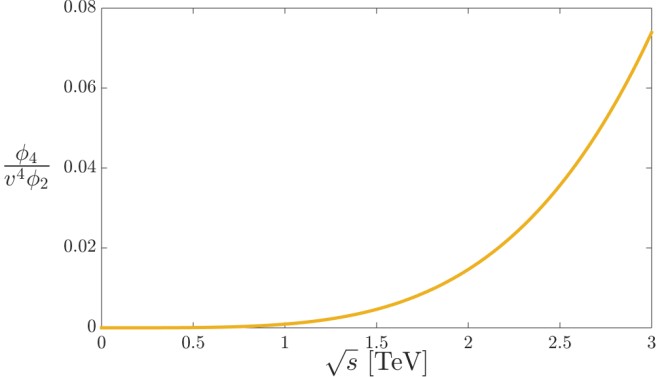

Figure 4: Ratio between the massless four-particle and two-particle phase spaces up to 3 TeV, that can be analytically evaluated from Eq. (46). Here $v = 246$ GeV.

Where $m_\rho = 770$ MeV. This is much smaller than the other terms of Eq. (18). For example, the first term $t_0 = \frac{s}{192\pi f_\pi^2}$ is about 0.12 at the $\rho$ meson pole; this amounts to a relative uncertainty from the correction at the level of four in ten thousand! Hence, the influence of the multi-pion cut in the $\rho$ pole position is very small.

This smallness is also reflected in the scaling of this source of uncertainty with the mass of the resonance. Taking the massless limit for simplicity for $\phi_4$ in Eq. (46) we see that $\phi_4/\phi_2 \sim s^2$, so that if $s^3/\pi |RC(t_1)|(s)$ scales as $s^2$ (a typical NLO ChPT contribution), then $\Delta(s)G(s)$ for the four-body intermediate contribution scales as $s^4$ (a three-loop effect). Therefore, this uncertainty to the mass of the resonances scales as $(m_{\text{resonance}}/\Lambda_U)^8$, with $\Lambda_U$ the

unitarity cutoff the theory, $4\pi f_\pi$ for hadron physics and $4\pi v$ for HEFT. Then, the influence of this uncertainty is very much diminished for $m_{\text{resonance}} < \Lambda_U$, though it raises strongly for higher energies.

## 4.4 From truncating perturbation theory for the elastic amplitude

Perturbative expansions, and among them Chiral Perturbation Theory, are organized as a geometric series homogeneous in an invariant energy squared $E^2$, that acts as a counting parameter. The naive uncertainty estimate upon truncating the series at order $n$ is of order $(E^2)^{n+1}$, which assumes that the typical coefficient multiplying this power is not excessively large, in order not to alter the relative geometric size of the term. This is suspected to fail, for example, when the number of allowed Feynman diagrams grows factorially, but it is widely accepted as the organizing principle and so we also adopt it (although a recent study [51] lays out an interesting probabilistic method to account for the uncertainty in the next order of such an expansion).

Increasing the order in the expansion of the EFT to improve the behavior of the unitarized method is a strategy employed, for example, in the $N/D$ method for $NN$ scattering [52]. Here we attempt to quickly estimate effects that appear at NNLO and have not been discussed in the other subsections of this section 4.

In the elastic Inverse Amplitude Method at NLO, two orders of the expansion have been kept. Can we make a statement about the third, neglected order, even if the method is not organized as a geometric expansion? If the next $t_2 \sim O(p^6)$ order of chiral-like perturbation theory for a given process and channel becomes known (which is the case in hadron physics [53,54], but not yet for the electroweak HEFT), one can expand the inverse amplitude to that higher order, using

$$G(s) = \frac{t_0^2}{t} \simeq \frac{t_0^2}{t_0 + t_1 + t_2} \tag{48}$$

and expanding to dispose of yet higher order contributions

$$G(s) = \frac{t_0^2}{t} \simeq t_0 - t_1 - t_2 + \frac{t_1^2}{t_0} \ . \tag{49}$$

(Some authors like thinking of the IAM as a Padé approximation, and at times criticize the inherent ambiguity in how to choose the right sequence of Padé approximants. It is the dispersive derivation that selects what is the appropriate sequence of approximations, by applying the EFT expansion to $G$.)

However, we wish to quantify how much can the pole get displaced by including the NNLO correction, not actually calculate to this order in every instance. This displacement can be captured by the low energy constants of the third order perturbation theory reflecting the imprint of a resonance on them. The relation between those constants and the resonance is known in Resonance Effective Theory [55–57](by integration of heavy degrees of freedom) and enters in the equation for the displacement of the pole in Eq. (21) as

$$
\begin{aligned}
3^{rd}PT &= G(0) + G'(0)s + \frac{1}{2}G''(0)s^2 - (t_0 - t_1)(0) + (t_0 - t_1)'(0)s + \frac{1}{2}(t_0 - t_1)''(0)s^2 \\
&= \left(\frac{t_1^2}{t_0} - t_2\right)(0) + \left(\frac{t_1^2}{t_0} - t_2\right)'(0)s + \frac{1}{2}\left(\frac{t_1^2}{t_0} - t_2\right)''(0)s^2 \ ,
\end{aligned} \tag{50}
$$

for the NNLO correction of $G$. (The derivatives at $s = 0$ exist even in the chiral limit as we are focusing on the polynomial contributions from the higher-order counterterms only.) This

Table 2: Renormalized scattering constants up to NNLO which we interpret as evaluated at the renormalization scale $\mu = \rho = 770$ MeV, this being the mass of the most prominent high-energy resonance eliminated to obtain them.

| Constants | Values consistent with [55, 58] |
|---|---|
| $(l_1, l_2, l_3)$ | $(2, 4, 10) \cdot 10^{-3}$ |
| $(r_1, r_2, r_3, r_4, r_5, r_6, r_F)$ | $(-17, 17, -4, 0.0, 0.9, 0.25, -1) \cdot 10^{-4}$ |

means that, for $IJ = 11$,

$$
\begin{aligned}
|\Delta(s)G(s)| = \frac{m_\pi^2}{480 f_\pi^6} \Big| &\big(4m_\pi^4(5r_2 + 20r_3 - 20r_4 + 72r_5 - 8r_6 - 10r_F) + \\
&+ m_\pi^2(-5r_2 - 40r_3 + 80r_4 - 216r_5 + 24r_6 + 10r_F)s + \\
&+ (-80l_1^2 + 80l_1 l_2 - 20l_2^2 + 5r_3 - 15r_4 + 54r_5 + 14r_6)s^2\big) \Big|.
\end{aligned}
\tag{51}
$$

Example values of the constants in Eq. (51) are given in Table 2.

Evaluating Eq. (51) at the $\rho$ mass, $s = m_\rho^2$, we find

$$
|\Delta(s)G(s)| \simeq 6 \cdot 10^{-3} .
\tag{52}
$$

Because the IAM matches the EFT at low $s$, $m_\pi$ to NLO order, we expect the uncertainty to scale with the typical NNLO $s^3$ behavior for (quasi)massless Goldstone bosons.

However, the explicit power of $m_\pi^2$ in Eq. (50) entails a $m_\pi^2 s^2$ contribution to the uncertainty (that we take to Table 3 shown later on).

### 4.5 From approximate left cut

The integral in the RHS of Eq. (15) is the most difficult piece of the IAM derivation to be bound or constrained, and it also contributes the largest uncertainty. We will divide this integral into different energy regimes distinguished by how the amplitude scales with energy. It will transpire that the largest contribution is due to the intermediate-energy part of the integration region, and since ChPT is expected to reasonably approximate the amplitude there, we will be able to bind the induced uncertainty.

In non-relativistic [59] theory, the left cut contains the information about the interaction potentials whereas the right cut contains the physical particles, so there is an intrinsic difference of treatment. In the relativistic theory on the other hand, the left cut of a given partial wave is related to the right cuts of other channels and partial waves due to crossing symmetry, as encoded for example in Roy-Steiner equations [1, 2].

If the integrand of Eq. (15) vanished, that is, $\text{Im}\,G = -\text{Im}\,t_1$, the left cut would receive exact treatment. This is obviously not the case, as $G$ is unknown *a priori*. Therefore, we examine the contributions of both $G$ and $t_1$ there.

#### 4.5.1 Inspection of the left-cut integral for $t_1(s)$

First, let us address $t_1$, *i.e.* the NLO term of the partial wave amplitude. To assess which part of the left cut integral is numerically most relevant we use, as a typical case, the $\mathcal{O}(p^4)$ ChPT partial wave amplitude for the $IJ = 11$ channel in the chiral limit from [60]. The low-$|z|$ part of the $LC(t_1)$ integral is suppressed by the derivative coupling in the effective theory, but the high-$|z|$ one is suppressed by the $z^3(z - s)$ factor from the dispersion relation. We should like

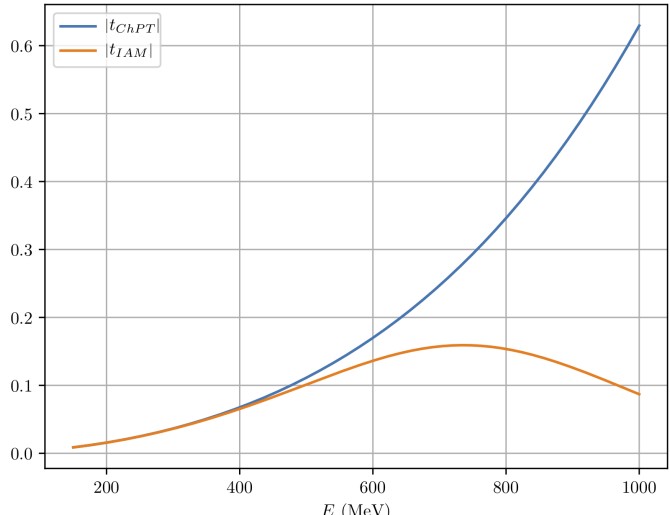

Figure 5: Comparison between the moduli of $t_{\text{ChPT}}$ and $t_{\text{IAM}}$ at $\mathcal{O}(p^4)$ for the $IJ = 11$ channel. The two amplitudes separate around a scale 200 MeV above the two pion threshold.

to see which one contributes the most, so we split the integration interval as

$$\frac{s^3}{\pi} LC(t_1) = \frac{s^3}{\pi} LC_{\text{far}}(t_1) + \frac{s^3}{\pi} LC_{\text{near}}(t_1) = \frac{s^3}{\pi} \int_{-\infty}^{-\lambda^2} ds' \frac{\text{Im } t_1(z)}{z^3(z-s)} + \frac{s^3}{\pi} \int_{-\lambda^2}^{0} ds' \frac{\text{Im } t_1(z)}{z^3(z-s)} , \quad (53)$$

with a contribution coming from a region near to the origin $s = 0$, $[-\lambda^2, 0]$, and a second from higher energies, $(-\infty, -\lambda^2]$. We choose $\lambda$ to be 470 MeV, i.e. the scale where the $\mathcal{O}(p^4)$ ChPT amplitude separates from the IAM amplitude and the scale where ChPT is supposed to give a good prediction of the full amplitude (200 MeV above the two pion threshold). This is presented in Fig. 5 where the moduli of $t_{IAM}(s)$ and $t_{\text{ChPT}}(s)$ are plotted along the RC.

Once $\lambda$ has been chosen, we compare the near and far (respect to the origin $s = 0$) left cut contributions to Eq. (53) (with $t_1$ in the integrand) in Fig. 6. Because of the structure of Eq. (11), we also include a line to compare $t_0 - t_1$. Note that around the resonance region the near left cut is about one order of magnitude larger than the far left cut.

The contribution from the near left cut of $t_1$ will be treated together with the integral of Im $G$ over the same region of integration. This is convenient since in the region $[-\lambda^2, 0]$ the function $G$ is well approximated by the NLO ChPT expansion and the combination Im $G$+Im $t_1$ is a NNLO remnant contribution. This is discussed in the next section.

Part of the contribution from the far left cut of $t_1$ to Eq. (53) will be treated together with a contribution coming from the $LC$ of $G$ in Eq. (63). The rest will be accounted for in Eq. (68).

### 4.5.2   Left Cut integral for $G + t_1$

We now turn to the part of the integrand in the RHS of Eq. (15).

For low-$|s|$, the quantity Im $(G + t_1)$ in Eq. (15) is controlled *a priori*, as perturbation theory is a good approximation to the scattering amplitude, and thus to $G$. This uncertainty is broadly included in the uncertainty counting of subsection 4.4.

At high-$|s|$ we can exploit the asymptotic behavior of the amplitude brought about by the generic relation between the left and the right cuts, described shortly, and in the case of an underlying non-Abelian gauge theory such as QCD, that is asymptotically free, the Brodsky-Farrar counting rules [61–64].

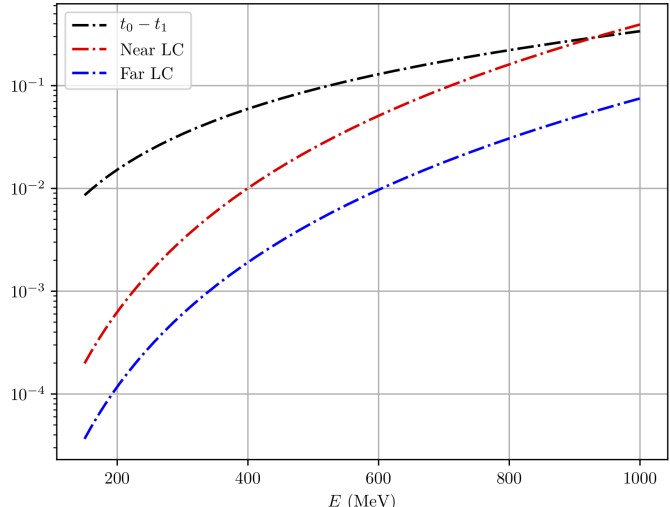

Figure 6: Comparison between the moduli of the near and far left cut pieces of Eq. (53) and $t_0 - t_1$ for the $IJ = 11$ channel.

This suggests that we can divide the integration domain and thus the integral for $G + t_1$ in the RHS of Eq. (15) as

$$\frac{s^3}{\pi} \int_{LC} dz \, \frac{\text{Im}(G + t_1)}{z^3(z-s)} \equiv I_1[G + t_1] + I_2[G + t_1] + I_3[G + t_1] \tag{54}$$

into three regions and discuss them separately. These will be an ultraviolet region extending through $(-\infty, -\Lambda^2)$, yielding $I_1$, an intermediate region $(-\Lambda^2, -\lambda^2)$ contributing to $I_2$, and the infrared region $(-\lambda^2, 0)$ that returns $I_3$. These $I_i$ are functionals of an amplitude-like function and functions of $s$ and their respective interval cutoffs.

**Low-$|z|$ region ($|I_3|$).** The lowest dividing scale, $\lambda$, is chosen to allow the use of ChPT for $|s| < \lambda^2$ since, as $|s| \to 0$,

$$\text{Im}\, G(s) = -\text{Im}\, t_1(s) + \mathcal{O}(s^3) \Rightarrow \text{Im}\, G(s) + \text{Im}\, t_1(s) = \mathcal{O}(s^3). \tag{55}$$

We naturally choose $\lambda$ to coincide with the value where we split the $LC(t_1)$ contribution in Eq. (53) ($\lambda = 470$ MeV). Automatically we can estimate $I_3$ from the near-to-the-origin contribution to $LC(t_1)$. That is, since the chiral counting is certainly valid up to this scale $\lambda$, our estimate to the error introduced by $G + t_1$ can be based on exposing the order of that counting,

$$|I_3[G + t_1](s, \lambda)| \simeq \left| \frac{s^3}{\pi} \int_{-\lambda^2}^{0} dz \, \frac{k|z^3|}{z^3(z-s)} \right| = \frac{ks^3}{\pi} \log\left(1 + \frac{\lambda^2}{s}\right), \tag{56}$$

where $k$ is a real positive constant encoding the third-order $\mathcal{O}\big((s/4\pi f_\pi^2)^3\big)$ remnant dynamical contributions. For example, the uncertainty introduced by this part of the left cut far in the right $s$-cut on the resonance region, $\lambda^2 \ll s \simeq M_R^2$, can be given as (note that $I_3[t_1] = \frac{s^3}{\pi} \times LC_{\text{near}}(t_1)(s)$)

$$|I_3[G + t_1]| = |I_3[G] + I_3[t_1]| \propto \frac{s^2 \lambda^2}{\pi(4\pi f_\pi)^6}, \tag{57}$$

which is indeed of the same order as would be suggested by ChPT itself, if unitarity played no role in the discussion.

With the counting $k \sim 1/(4\pi f_\pi)^6$, in hadron physics $\sqrt{s} \sim 0.77$ GeV, $f_\pi \sim 0.092$ GeV and $\lambda \sim 0.47$ GeV, this treatment leads to a very acceptable uncertainty $I_3 \sim 1\%$. In HEFT this is presumably similar upon rescaling $\lambda$ and $s$ to account for $v = 246$ GeV instead of $f_\pi$.

In spite of $LC_{near} > LC_{far}$ when $t_1$ alone is considered, more generally, because of the near cancellation of Im $G = -$Im $t_1$, we find the low-$|z|$ contribution to be rather small.

**High-$|z|$ region ($|I_1|$).** The larger dividing scale $\Lambda$, defining region 1, is chosen to be able to invoke the Sugawara-Kanazawa (SK) theorem [65]. This theorem may be useful because it relates the asymptotic behavior of the right cut (dominated by unitarity, or further out by Regge theory, and in any case constrained by experimental data) to that of the left cut (about which much less is known and that we wish to constrain).

To be precise, the theorem states that if the function $f(z)$ has the same analytic structure as the partial waves in Fig. 1; if it diverges at most as a finite power of $z$ as $|z| \to \infty$; if it has finite limits in the RC as $z \to \infty \pm i\epsilon$, $f(\infty \pm i\epsilon)$, and the limits at infinity in the LC exist, then

$$\lim_{|z| \to \infty} f(z) = f(\infty + i\epsilon) \quad \text{Im } z > 0,$$
$$\lim_{|z| \to \infty} f(z) = f(\infty - i\epsilon) \quad \text{Im } z < 0. \tag{58}$$

For example, the direct application of the SK theorem to the scattering amplitude, satisfying Eq. (4) (unitarity bound $|t| \le 1/\sigma$) would entail that $\lim_{|z| \to \infty} t(z)/z = 0$. However, it is not clear that the second of the hypotheses of the theorem, polynomial boundedness, is satisfied by partial waves in the scattering on composite objects [27], since an exponential growth is expected for large imaginary $z$. Luckily, for the inverse amplitude the exponential factor is actually damping. So that, for composite objects, $G$ vanishes as Im $z \to \infty$.

Indeed, our interest is to apply the SK theorem directly to $G$. In the entire construction of the IAM, from Eq. (10) on, we have assumed that $t(s)$ does not decrease along the right cut faster than or proportionally to $1/s$, since we subtracted $G$ three times (there is no obstacle in increasing the number of subtractions if $t$ decreases at large energy, but Regge theory suggests that this is not the case). Actually, it is known that Regge theory gives a scaling of $t(s) \propto$ constant $\neq 0$ in the limit of large $s$ over the right cut for the $IJ = 11$ channel [40].

Standard deployment of the SK theorem also requires that any poles of $G(s)$ should lie between the cuts: however, the function $G(s)$ could present CDD poles along the cuts. If their number $N_C$ is *finite*, after identifying them we can still use the theorem for $G$ by constructing an auxiliary function of complex $s$ where the poles have been shifted to an acceptable position,

$$\widetilde{G}(z) = G(z) \prod_{j=1}^{N_C} \frac{z - s_j}{z - \epsilon_j}. \tag{59}$$

There, $s_j$ are the original positions of the CDD poles of $G$ along the cuts and $\epsilon_j$ are the positions of an equal number of added poles that lie between the cuts, so that they are accounted for by the SK theorem in standard form. In this way, the auxiliary function $\widetilde{G}(z)$ has no CDD poles along the cuts and, because of the SK theorem we then conclude that for $z \to -\infty \pm i\epsilon$ it has the same limit as in the RHC for $z \to +\infty \pm i\epsilon$, respectively.

Then, the function $G$ diverges as $s^2$ and for fixed $s$ is bounded in magnitude by $k's^2 \propto s^2/(4\pi f_\pi)^4$. Similarly, we also know that $G$ is bounded by a constant times $|z|^2$ for $z \to \infty$ in any direction.

As a result of this analysis on the asymptotic behavior of $G(z)$, we conclude that in the high energy regime we have the following bound on $|I_1[G]|$,

$$|I_1[G](s,\Lambda)| \equiv \frac{s^3}{\pi}\left|\int_{-\infty}^{-\Lambda^2}dz\frac{\operatorname{Im}G(z)}{z^3(z-s)}\right| \leq \frac{k's^2}{\pi}\log\left(1+\frac{s}{\Lambda^2}\right). \tag{60}$$

To estimate the logarithm we choose $\Lambda = 1.4\,\text{GeV} \simeq \sqrt{2\,\text{GeV}^2}$. This reasonable threshold for the division of intervals was used in earlier literature [66, 67] to match the phase of the $\pi\pi$ scalar form factor with its smooth asymptotic expression given by the Brodsky-Farrar quark-counting rules [61–64] for $s > \Lambda^2$, with good phenomenological success. Although these counting rules do not strictly apply to partial-wave amplitudes, that pick up Regge contributions due to the angular integration, (semi-local) duality also suggests that intermediate-energy resonances' contributions would average out already for $s \lesssim \Lambda^2$ to the smooth (Regge) asymptotic expression. Indeed Ref. [40] also argues that above energies $|s|^{1/2}$ between 1.3 to 2 GeV, either perturbative QCD -in fixed angle problems- or Regge theory -for partial wave amplitudes- are applicable.

An estimate of $k'$ in Eq. (60) can be obtained by matching the inferred asymptotic behavior of $\operatorname{Im}G$ along the LC with an estimate of its value at $s = -\Lambda^2$ ($|\operatorname{Im}G(-\Lambda^2)| < k'\Lambda^4$). We use the following approximation,

$$k' \sim \frac{|\operatorname{Im}G(-\Lambda^2)|}{\Lambda^4} \to \frac{|\operatorname{Im}t_1(-\Lambda^2)|}{\Lambda^4}, \tag{61}$$

where we have taken into account that both $t_1$ and $G$ have the same $\propto s^2$ asymptotic behavior for $s \lesssim -\Lambda^2$, and we employ the former to ascertain the order of magnitude for $k'$.

This estimate of $k'$, once more in our reference scenario of ChPT and at the $\rho$ meson scale, becomes $k' \simeq 0.15$ at $\Lambda$. This figure can of course be changed by shifting $\Lambda$, passing to or bringing contributions from the intermediate energy region in the next paragraph. This yields a variation of 0.05 above or below, taking $s$ in the range $(-2\,\text{GeV}^2, -1\,\text{GeV}^2)$. Eq. (60) then yields $|I_1[G]| \sim 0.01$ and we conclude that $|I_1[G]|$ is roughly $\mathcal{O}(1\%)$

We now combine Eq. (60) with an equivalent piece of the left cut carrying $t_1$,

$$(s^3/\pi)LC_{\text{far}}(t_1) \equiv I_1[t_1](s,\Lambda). \tag{62}$$

Since for $I_1[G]$ we only estimated a quote for its modulus we add it in quadrature with $I_1[t_1]$, so that

$$|I_1[G+t_1](s,\Lambda)| \lesssim \left(\left[\frac{k's^2}{\pi}\log\left(1+\frac{s}{\Lambda^2}\right)\right]^2 + \frac{s^6}{\pi^2}\left|\int_{-\infty}^{-\Lambda^2}dz\frac{\operatorname{Im}t_1(z)}{z^3(z-s)}\right|^2\right)^{1/2}. \tag{63}$$

$|I_1[t_1]|$ is evaluated, in ChPT and for the $IJ = 11$ channel ($s = (0.77\,\text{GeV})^2$), to be $4.6 \cdot 10^{-3}$. Altogether, we have from Eq. (63) $|I_1[G+t_1]| \sim 0.01$.

**Intermediate-$|z|$ region.**($|I_2|$) This is the energy range that we have found most difficult to discuss, as the amplitude has no simple power-law behavior. We have attempted to lean on Mandelstam and Chew's work to express, through crossing, $\operatorname{Im}t(s)$ over the left cut as a combination of partial waves over right cuts in different $IJ$ channels (see Eq (IV.6) in [9]). However, this characterization is not easily controllable for numerical computations. Therefore, we show three possible simpler estimates that are broadly consistent.

For the first estimate we change in the quantity to be reduced,

$$I_2[G+t_1](s,\Lambda,\lambda) \equiv \frac{s^3}{\pi}\int_{-\Lambda^2}^{-\lambda^2}dz\frac{\operatorname{Im}(G+t_1)}{z^3(z-s)}, \tag{64}$$

the integral over $G + t_1$ by the integral over $t_1$ as a presumed upper bound, observing that they partially cancel each other (since in the range of lowest energies involved $\text{Im} G = -\text{Im} t_1 + \mathcal{O}(s^3)$) and that $G$ is expected to be of the same order than $t_1$. We would have the bound

$$I_2[G + t_1](s, \Lambda, \lambda) < |I_2[t_1](s, \Lambda, \lambda)| . \tag{65}$$

This is the weakest point of the reasoning and could fail should $G$ turn out much stronger than $t_1$, which can happen for example if a zero of $t$ resides in the intermediate left cut. In such a case, one should isolate it by considering $\text{Im}\,(G + t_1)(s - s_{\text{zero}})/(s - s_1)$', with $0 < s_1 < 4m_\pi^2$. We ignore how likely this is, so that in a practical uncertainty analysis it needs to be checked case by case employing Eq. (26). With the caveat, this should be sufficient for an uncertainty estimate with Eq. (65) ascertaining its typical order of magnitude.

Evaluating again Eq. (65) for $s = (0.77 \text{ GeV})^2$ in ChPT for the $IJ = 11$, we see easily that $|I_2[G + t_1]| \lesssim 6 \cdot 10^{-2} \simeq 6\%$. This would make it the biggest contribution to the uncertainty of all those examined, deserving further scrutiny.

We now follow a second, different path to $I_2[G + t_1](s, \Lambda, \lambda)$ so as to estimate the typical size expected for this contribution $I_2$, which avoids the discomfort of assuming that $\text{Im} G$ and $\text{Im} t_1$ are of similar magnitude. Our point now is that higher derivatives respect to $s$ of the integral in Eq. (64) are dominated by the integrand near the upper integration limit $z \lesssim -\lambda^2$ for positive values of $s \ll \Lambda^2$.

To arrive to this result analytically, and for this purpose only, we take the low-energy expression $\text{Im}(G + t_1) = k s^3 = \mathcal{O}(s^3)$ with $k \simeq 1/(4\pi f_\pi)^6$, which is valid for $z \lesssim -\lambda^2$ in the upper part of the integrand. Thus,

$$\int_{-\Lambda^2}^{-\lambda^2} dz \frac{\text{Im}\,(G + t_1)}{z^3(z - s)} \to k \log \frac{s + \Lambda^2}{s + \lambda^2} . \tag{66}$$

The $n_{\text{th}}$ order derivative of the RHS is

$$(-1)^{n-1}(n-1)! \left( \frac{1}{(s + \lambda^2)^n} - \frac{1}{(s + \Lambda^2)^n} \right) \approx \frac{(-1)^{n-1}(n-1)!}{(s + \lambda^2)^n} , \tag{67}$$

where the last step is valid for $s$, $\lambda^2 \ll \Lambda^2$, being quantitatively sensible already for $n = 1$. However, its primitive given by the RHS of Eq. (66) reflects the assumption done for the higher-energy region of $\text{Im}(G + t_1)$. The general form for the primitive of the last term in Eq. (67) for $n = 1$ is $\log(s + \lambda^2) + C$, where $C$ is a constant that, because of dimensional analysis, should be written as $-\log(L^2)$. Here $L$ is a hard scale reflecting the deeper energy tail of the integration interval. By estimating it we compute $C$, which can then be considered as a counterterm that we calculate in terms of a natural-size scale $L$. For numerical computations we allow $L^2$ to take values between $\Lambda^2 = 2 \text{ GeV}^2$ up to $2\Lambda^2 = 4 \text{ GeV}^2$. The latter is motivated because in many QCD sum rules the onset of perturbative QCD input for the spectral functions is precisely taken at 4 GeV² [68–70]. Then we have the following expression for $I_2[G + t_1]$ in Eq. (64) for positive $s$, $\lambda^2 \ll \Lambda^2$,

$$I_2[G + t_1](s, \Lambda, \lambda) = \frac{s^3 k}{\pi} \log \frac{s + \lambda^2}{L^2} , \ L^2 \in [2, 4] \text{ GeV}^2 . \tag{68}$$

Evaluating Eq. (68) at $s = (0.77 \text{ GeV})^2$, we have $|I_2[G + t_1]|$ ranging between 1.7-3.5%. As a check of consistency, we would like to mention that our estimate for $|I_2[G + t_1]|$ is consistent with the more reliable one for $|I_1[G + t_1]|$. Taking into account the scales driving both contributions, one would expect that $|I_2| \sim |I_1| \Lambda^4/(4\pi f_\pi)^4 \simeq 2.4\%$, which is well inside our estimate for $|I_2|$.

Those two estimates can be deployed for HEFT as soon as data beyond the Standard Model is available. But in hadron physics we can perform a third, independent check with an explicit calculation of this intermediate energy region in Eq. (64).

We have used, for $I = J = 1$, the amplitude $t(s)$ from Ref. [43],[8] where the $\pi\pi$ $S$ and $P$ waves are parameterized analytically *in the physical region* based on the the GKPY equation [2] and reproduce $\pi\pi$ scattering data up to $\sqrt{s} = 2$ GeV. The integral $|I_2[G+t_1]|$ evaluates to 0.08, of the same size but on the larger side of the first estimate. The reason is that this channel is close to the worst scenario because the $\pi\pi$ $P$ wave, as calculated from Ref. [43], becomes very small around $-0.36$ GeV$^2$, so that $|\mathrm{Im}\, G|$ is much larger in this region than the typical values for any of the $S$ waves from the same reference. This is due to the presence of zeroes in both the real and imaginary parts of the $I = J = 1$ partial-wave amplitude that are almost coincident in energy. The exact relative difference between these zeroes strongly affects the result of $I_2[G]$. Therefore, an error estimate would be desirable (if baroque: an uncertainty on the uncertainty!) to provide a numerically accurate calculation of $I_2[G+t_1]$ for this partial-wave amplitude, beyond estimating its order of magnitude as done here. Unfortunately, the uncertainty in the parameterization of [43] is presently not known for the left cut. Nonetheless, this estimate is conservative enough to fairly account for the magnitude.

**Total Left Cut uncertainty.** We finally put together the three pieces $I_i[G+t_1]$ from Eq. (56), (63) and (68) yielding the uncertainty to Eq. (15),

$$\Delta(s)G(s) = \sum_i I_i[G+t_1] = \sum_i I_i[G] + \sum_i I_i[t_1]. \tag{69}$$

Proceeding to the quantity controlling the displacement of the pole in Eq. (18), we add in quadrature the different sources of uncertainties which moduli have been bounded previously. Attending to the estimated typical size for them we have

$$|\Delta(s)G(s)| \le \sqrt{0.01^2 + 0.03^2 + 0.01^2} \simeq 0.04\,. \tag{70}$$

This is then propagated to Eq. (18) for estimating the displacement of the pole (see also Eq. (81) in Appendix A.1 below). The resulting typical size expected for the uncertainty in the pole position of the $\rho(770)$ is around a 17%, which is given also in the last line of Table 3.

In the extreme case that we use the parametrization for the $I = J = 1$ $\pi\pi$ partial-wave amplitude provided in Ref. [43] we have the larger uncertainty

$$|\Delta(s)G(s)| \le \sqrt{0.01^2 + 0.08^2 + 0.01^2} \simeq 0.08\,, \tag{71}$$

which would double the value of the relative error in the pole displacement up to 34%. However, this calculation is very sensitive to the *exact* position of the zeroes in the real and imaginary parts of the partial-wave amplitude, whose uncertainty in position is not provided.

*A posteriori* in hadron physics we do of course know that the IAM *predicts* the $\rho(770)$ at 710 MeV with the central values from threshold determination of the ChPT low energy constants [71], so that the error is of order 10% and not 30%.

This left-cut uncertainty is by far the largest entry in the table that can only be improved partially by including higher-orders in the chiral series. This is because the largest uncertainty stems from the intermediate-energy region along the LHC, $I_2[G+t_1]$, which could be further suppressed, albeit only slowly, by including more subtractions since $m_\rho \lesssim \Lambda \sim 4\pi f_\pi$, being the latter the natural scale in ChPT suppressing loop contributions (like those generating $I_2$). Nonetheless, a better knowledge of this part of the LC could sharpen our estimate of the uncertainty. This improvement would be likely associated with a better unitarization method compared to the NLO IAM.

---

[8]We thank Jacobo Ruiz de Elvira for providing the data.

### 4.6 A comment on crossing symmetry violation

While the Effective Theory may be crossing symmetric, as it is a Lorentz-invariant local field theory, the unitarization of its amplitudes is a procedure that, as we have seen, treats the left and right cuts in a different manner. This leads to questions about to what extent can crossing symmetry be respected [72, 73].

Crossing symmetry is manifest when comparing the three isospin amplitudes in $SU(2)$ Goldstone boson scattering,

$$
\begin{aligned}
T_0(s,t) &= 3A(s,t,u) + A(t,s,u) + A(u,t,s), \\
T_1(s,t) &= 3A(t,s,u) - A(u,t,s), \\
T_2(s,t) &= A(t,s,u) + A(u,t,s),
\end{aligned}
\tag{72}
$$

all three of which are written in terms of only one function $A(s,t,u)$ analytically extended to different Mandelstam kinematic regions and with the arguments swapped.

If $A(s,t,u)$, $A(t,s,u)$ and $A(u,t,s)$ were three independent complex functions $A$, $B$ and $C$, then they could be reconstructed by separately applying the IAM to each partial wave of fixed isospin, and employing the partial-wave projection in each channel to reconstruct the three $T_0$, $T_1$, $T_2$, then solving the linear system in Eq. (72) to obtain all three. But the fact that $A$, $B$, $C$ are analytic extensions of each other with the arguments swapped leads to subtle relations between them.

A practical way to expose them is to encode the information in integral relations between the partial wave amplitudes, the Roy equations [74]. However, these equations relate partial waves with different angular momenta and the number of computations accessible to the NLO IAM is limited to $J < 2$. Therefore, neither a reconstruction of $A$ nor a test of the Roy equations is really sensible: the IAM at NLO is not predictive enough to be tested by crossing symmetry. What it does, parametrize a few low-lying partial waves into the resonance region, it does reliably enough as we have exposed through the work, but its reach is not sufficiently extended to make a statement about crossing.

Nevertheless, a direct attempt to test crossing in the physical region from Eq. (72) found sizeable violations [72]; But any reasonable parametrization of the data, not only the IAM, will find very similar results. Either crossing cannot be precisely tested without higher angular momentum waves, or the data points themselves are violating crossing symmetry (!).

Such attempt at testing crossing with the IAM at very low $s$ where only the first partial waves contribute to the amplitude [72] employs the Roskies relations [75]. These are integral relations for the amplitudes between $s = 0$ and threshold. The first few relations are very well satisfied [76] to $\mathcal{O}(1\%)$, and the agreement only deteriorates upon increasing their order. But this is a region where the IAM (once the Adler zero is taken care of: the mIAM was not known in 2001, but already then an approximate subtraction of the subthreshold zeroes was carried out [72]) essentially coincides with chiral perturbation theory, so that the test should be adscribed to the uncertainty of the EFT itself, and not to that of the unitarization method.

## 5 Conclusion and outlook

### 5.1 Summary of systematic uncertainty sources

It is looking increasingly likely that the LHC will not find new resonances, but it may still have a chance of revealing non-resonant [77] separations from Standard Model cross-sections, particularly in the Electroweak Symmetry Breaking Sector formed by the Higgs scalar boson $h$ and longitudinal electroweak bosons $\omega$.

Table 3: Different sources of uncertainty for the Inverse Amplitude Method in Hadron Physics, their scaling if relevant, the order of magnitude of the error they introduce in the resonance region and whether the basic method can be improved to remove that source of uncertainty if need be.

| Source of uncertainty | Equation | Behavior | Pole displacement at $\sqrt{s} = m_\rho$ | Can it be improved? |
|---|---|---|---|---|
| Adler zeroes of $t$ | (25) | $(m_\pi/m_\rho)^4$ | $10^{-3}$ - $10^{-4}$ | Yes: mIAM |
| CDD poles at $M_0$ | (29) | $M_R^2/M_0^2$ | $0$ - $\mathcal{O}(1)$ | Yes: extract zero |
| Inelastic 2-body | (37) | $(m_\rho/f_\pi)^4$ | $10^{-3}$ | Yes: matrix form |
| Inelastic 4...-body | (47) | $(m_\rho/f_\pi)^8$ | $10^{-4}$ | Partially |
| $O(p^4)$ truncation | (51) | $(m_\pi^2 m_\rho^4)/f_\pi^6$ | $10^{-2}$ | Yes: $O(p^6)$ IAM |
| Approximate Left Cut | (70) | $(m_\rho/f_\pi)^6$ | $0.17$ | Partially |

If that is the case, extrapolation of the data to higher energies will be needed to determine the new physics scale. One tool to do this is unitarization of the low-energy EFT, and controlling the uncertainties in the method is therefore necessary. Among the many versions of unitarization, those that have an underlying derivation in terms of a dispersion relation are more amenable to controlled systematics on the theory side. We have examined the Inverse Amplitude Method, salient among unitarization procedures and well understood in theoretical hadron physics, but whose uncertainties have not yet been systematically listed.

Table 3 summarizes the sources of uncertainty that we have identified and the status of each (whether it is or not amenable to systematic improvement).

In the table we have explicitly spelled out our knowledge of the scaling with energy of the various contributions to the uncertainty. If instead of the $\rho(770\,\text{MeV})$ in QCD, or an analogous particle with proportional mass (*e.g.* $Z'$) in EW model extensions, we had taken a resonance with a larger mass (above that unitarity scale $\Lambda_U$ of $4\pi f_\pi$ in QCD, or the generic $4\pi v$ in a BSM theory) the entries in table 3 labelled "Partially" would become a definite "No". For the remaining entries one could set in the estimates the mass of the resonance around 1.5 GeV (around twice the $\rho$ mass) and calculate the numbers.

For resonances originating below the fundamental cutoff, such as the renowned $\sigma/f_0(500)$ meson of QCD, the estimated uncertainty would be even smaller than found here (for the $\rho(770)$), even though it is a wide resonance. Specifically, the estimate of the largest uncertainty source along the LC coming from the intermediate energy region in Eq. (68) scales as $s^3$. Because $|s_\sigma/M_\rho^2|^3 \approx 0.11 \ll 1$, with $s_\sigma$ the pole position of the $\sigma/f_0(500)$ resonance [78], the uncertainty is indeed much smaller.

It is true that dealing with such a broad resonance could modify the straightforward applicability of Eq.(16) to propagate the error bar to the pole position. But even then, we think that the calculated uncertainties would be sensible at the semiquantitatively level at least. In any case, one could exquisitely perform the error propagation by attending directly to the resonance pole position in the second Riemann sheet by extending Eq. (18).

Thus, as long as we are below the unitarity scale $4\pi f_\pi$ (or $4\pi v$), the scaling of the uncertainty can be estimated directly from table 3, although the numerical coefficient of that scaling does depend on the $IJ$ channel.

The dominant $(m_{\text{resonance}}/f_\pi)^6 \sim (m_V/v)^6$ dependence (coming from the left cut uncertainty) eventually becomes of order 1 and the method is overpowered by the error. It then loses its applicability in its known form as presented here. In QCD this would happen when $1 \sim 0.17 \times (m_{\text{max}}/m_V)^6$, *i.e.*, for $m_V \simeq 770$ MeV, $m_{\text{max}} \simeq 1050$ MeV (that is in the ballpark of $4\pi f_\pi$ where the entire EFT setup becomes dubious anyway).

Also interesting is the scaling of the low-energy constants [55] in table 2 that we have used to estimate the uncertainty in the truncation at a given perturbative order (NLO). The NNLO constants governing the uncertainty behave as $f_\pi^4/m_V^4$, which is a very steep dependence: the

NLO $l_i \propto f^2/m_V^2$ see a slower fall off and therefore are more important at low energy. The NNLO constants tend to become less visible as the mass of the resonance increases.

And last, the inelastic four-Goldstone boson channel becomes of the same order as the two-Goldstone boson channel at about the same energy $\Lambda_U$ (1.2 GeV for hadron physics) as inferred from our computation of the relevant phase space in subsection 4.3. An extension of the IAM to a more complicated method is then mandatory.

## 5.2 Comparison to statistical uncertainties from eventual measurements

We have dealt with theoretical systematic uncertainties. But eventual experimental measurements of the low-energy constants will have an uncertainty, presumably very dominated by statistical fluctuations at initial stages.

The question of interest here is to what level do these uncertainties on the parameters of the low energy theory have to be pushed down so that the theory uncertainties in this article become the pressing issue. This is exemplified in figure 7.

The plot shows a computed elastic $\omega_L \omega_L$ vector resonance chosen to show up at 3.8 TeV, approximately. A 17%-sized uncertainty band, as suggested by the largest entry in Table 3, would make the resonance mass uncertain between about 3.2 and 4.4 TeV. This may seem like a large uncertainty, but note that in the graph there are additional plots obtained by varying the low-energy constants that govern this $J = 1 = I$ channel. The variation levels that are roughly equivalent to the IAM systematics are

$$\frac{\Delta(1-a)}{(1-a)} \simeq \left\{ \begin{array}{c} +30\% \\ -40\% \end{array} \right\} \; ; \qquad \frac{\Delta(a_4-2a_5)}{(a_4-2a_5)} \simeq \left\{ \begin{array}{c} +50\% \\ -30\% \end{array} \right\} . \tag{73}$$

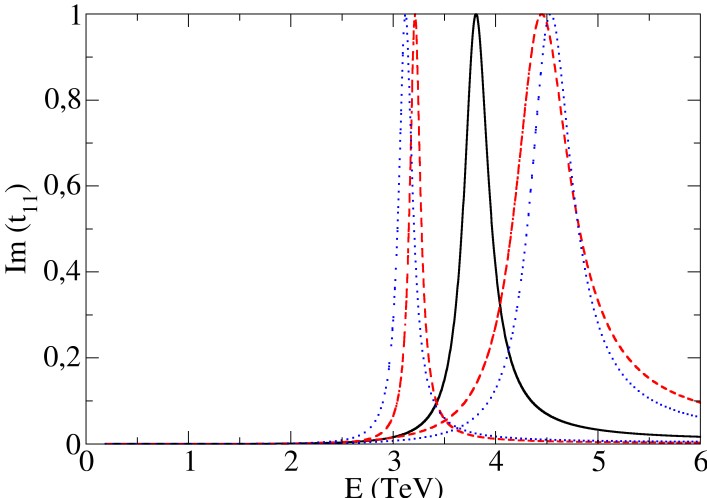

Figure 7: Solid line in the middle: vector-isovector resonance generated with the HEFT Lagrangian with parameters $a = 0.95$ and $(a_4 - 2a_5) = 10^{-4}$ at $\mu = 3$ TeV. The mass of the resonance turns out to be about 3.8 GeV. Dashed lines to either side (red online): recalculated resonance with $a = 0.965$ and $a = 0.93$. Dotted lines (blue online): alternatively, $a$ is fixed at 0.95 and $a_4 - 2a_5$ increased by 50% or decreased by 30%, respectively, with similar result. (The larger the separation from the Standard Model values $a = 1$, $a_4 - 2a_5 = 0$, the further to the left).

This means that, before the IAM systematic uncertainty became dominant, the measurement of the LO BSM coefficient $(1-a)$ as well as the NLO coefficient $(a_4 - 2a_5)$ would need to be accurate at a level of 30%. Presently, $a$ is compatible with 1 at the 10% level and no deviation from the SM is seen at LO; but how soon that 30% might be achieved for the NLO combination cannot be easily answered.

This comparison would have to be carried out in each particular application of the analysis for a given experimental channel.

## 5.3 Constraining the "new physics" from alternative channels

It is possible that whatever extant new or higher scale physics manifests itself at low energy not obviously by the elastic scattering of the (pseudo)Goldstone bosons but by their production from any other particle set. For example, in the EWSM we have coupled the $\omega\omega$ sector to the top-antitop quark channel [79] or to the $\gamma\gamma$ diphoton one [80].

The important point to remember is that Watson's final state theorem applies in the presence of strongly interacting new physics (as corresponds to a unitarity-saturating resonance). Then, the phases of the production and the elastic scattering amplitudes coincide. For example, we can use the following IAM approximation to the vector form factor that would control production of a vector resonance,

$$F_{11}(s) := \frac{t_{11}^{(0)}(s)}{t_{11}^{(0)}(s) - t_{11}^{(1)}(s)} \ . \tag{74}$$

This formula [81], obviously inspired in the IAM of Eq. (12) satisfies $F_{11}(0) = 1$, has the right cut due to unitarity, agrees with the perturbative expansion of the form factor at NLO, and most importantly, has the same resonance pole at the same position as the elastic amplitude. Thus, the uncertainty analysis performed to constrain that position in the later is carried to the former. Of course, such model introduces additional theoretical uncertainty, but this is in the *shape* of the form factor, not in the position of its resonance.

One can construct more reliable models that reduce even that uncertainty in the shape at the price of introducing an integral formulation. For example, the Omnès representation (that we have used among other things for the scalar form factor [41]) expresses it in terms of the phase shift inherent to the amplitude in Eq. (12), $t = \sin \delta e^{i\delta}$,

$$F_{00}(t) = P(t) exp \left( \frac{t}{\pi} \int_{4m^2}^{\infty} \frac{\delta_{00}(s)}{s(s - t - i\epsilon)} \right) , \tag{75}$$

with $P(t) = P_0 \prod_i (t - t_i)$ a polynomial constructed from the zeroes of $t$. While the modulus of the form factor is involved, its phase is directly $\delta_{00}(s)$ in agreement with Watson's theorem (it suffices to take the imaginary part of the integral in the exponent to see it). Thus, the Omnès construction allows to assign the very same uncertainty of the phase of the elastic partial wave to the phase of the form factor. This is all that is needed to have predictive power about the position of new physics. On the contrary, the intensity of its production in a given process (needing the modulus of the production amplitude) is itself afflicted by new sources of uncertainty that we do not consider here.

## 5.4 Conclusion

We hope to have provided a good stride in systematizing the study of its uncertainties and that it may one day be useful at the LHC and inspire further studies for this or other unitarization alleys. Our numerical estimates have most often relied on known hadron-physics parameters

and phenomena, but they are expected to be similar in Higgs Effective Field Theory beyond the Standard Model (because the interplay between derivative couplings and unitarity is universal once the correct scale has been fixed) so they can be used as a rough guide in BSM searches if any HEFT parameter separates from the Standard Model.

*A posteriori*, we know that the IAM in hadron physics yields, in the channel where we have concentrated most, the vector-isovector $\rho$ resonance, a calculated mass $m_\rho \simeq 710$ MeV instead of 770 MeV [71], when using the central values of the chiral counterterms fitted to low-energy data with ChPT. This is an error of order 8%, that is way better than the 20% uncertainty that we have estimated *a priori* for the largest source thereof, and have given in Table 3.

We have not shown details of the well-known scalar-isoscalar channel, but IAM computations in the 1990's [18] already predicted a $\sigma$-meson pole at $(440 - i245)$ MeV, whereas the detailed and precise extraction from Roy equations is yielding $(450 \pm 20) - i(275 \pm 10)$ MeV, approximately. Thus, the IAM seems to be faring even better than we have a right to state with the uncertainty analysis that we have carried out, and since this is dominated by the left-cut, it is likely that our bounds can be improved in the future.

Very clearly, what we estimate to be the major source of uncertainty, which should not surprise practitioners in this field, is the approximated left cut, which we have the most trouble constraining and easily brings a 10% loss of precision in extrapolation to high energy. But this should be sufficient, if separations from the SM are identified, to ascertain what energy scale should a future collider reach to be able to study the underlying new physics.

As for the common usage of the Inverse Amplitude Method, we should also emphasize that, prior to a search for resonance poles, one needs to examine the amplitude (through its chiral expansion) for the possible presence of zeroes (see subsection 4.1). If one is present in the resonance region, typically driving the partial-wave amplitude to having a narrow resonance with a zero close to its mass, the expansion and construction of the IAM have to be modified to account for it. We have introduced here the necessary methodological modification for such circumstances.

# Acknowledgements

We thank Jose R. Pelaez and Juan J. Cillero for several discussions, as well as the assistance of Ana Martín Fernández at early stages of the investigation, and J. Ruiz de Elvira for providing us with numerical data from the Roy equations.

**Funding information** This publication is supported by EU Horizon 2020 research and innovation programme, STRONG-2020 project, under grant agreement No 824093; grants MINECO:FPA2016-75654-C2-1-P, MICINN: PID2019-108655GB-I00, PID2019-106080GB-C21 (Spain); Universidad Complutense de Madrid under research group 910309 and the IPARCOS institute; and the VBSCan COST Action CA16108. JAO would like to acknowledge partial finantial support by the grants FEDER (EU) and MINECO (Spain) FPA2016-77313-P and MICINN AEI (Spain) PID2019-106080GB-C222/AEI/10.13039/501100011033.

# A Appendix

## A.1 Behavior and scaling of pole position uncertainty

Complementing subsection 4.2 we here present a simplified analytical derivation of the behavior of the pole position and its relative displacement due to the two kaon inelastic channel. These simple results are consistent with a numerical check on the displacement of the $\rho$.

In the chiral limit, the real part of the standard IAM equation for the pole position, $s_R$, Eq. (16) can be parametrized as

$$as_R - bs_R^2 = 0 . \tag{76}$$

(The logarithm multiplying the NLO $s^2$ piece, subleading to the power, need not be kept for the purpose of estimating the uncertainty.) This is solved by $s_R = a/b$. When the two body coupled channel correction in (37) is included we have to modify Eq. (76), using Eq. (18), as

$$as_R - bs_R^2 = cs_R^2 . \tag{77}$$

Assuming the correction is sufficiently small, the pole displacement is

$$s_R \to \frac{a}{b}\Big(\frac{1}{1+\frac{c}{b}}\Big) \simeq s_R\Big(1-\frac{c}{b}\Big) = s_R\Big(1-s_R\frac{c}{a}\Big) . \tag{78}$$

The constant $a$ can be taken from LO ChPT, for example, in the vector channel, $t_0 = as \Rightarrow a = -1/96\pi f_\pi^2 \simeq 0.4\,\text{GeV}^{-2}$, and to estimate the correction term $c$ (sloppily, since this was an upper bound and we ignore the expression's sign) $cs_R^2 \sim \Delta(s_R)G(s_R) = 0.08\frac{s_R^3}{\pi}RC(t_1)(s_R) \Rightarrow c \simeq 2\cdot 10^{-4}\text{GeV}^{-4}$. This means that we can compute the displacement of the pole in Eq. (78)

$$s_R\frac{c}{a} \simeq 0.001 . \tag{79}$$

So that the $\rho$ pole is uncertain at order $10^{-3}$ by a possible kaon coupled-channel induced displacement. The behavior with energy of this uncertainty is readily obtained

$$|\Delta(s)G(s)| = |cs^2| \simeq 5\cdot 10^{-4}\Big(\frac{\sqrt{s}}{4\pi f_\pi}\Big)^4 . \tag{80}$$

This treatment can be immediately extended to the uncertainties coming from four particle coupled channels and from the next order in perturbation theory in the EFT, as both sources scale as $s^2$. For the first we obtain a displacement of the pole of order $10^{-4}$. The second source of uncertainty affects the pole's position at order $10^{-2}$.

The largest error in the budget comes from the left cut. We have that $cs_R^3 \sim 0.04$, hence the pole gets displaced

$$s_R^2\frac{c}{a} \simeq 0.17 . \tag{81}$$

So that the pole may get displaced at order $10^0$ upon approximating the left cut. The energy behavior of this uncertainty is $(m_\rho/f_\pi)^6$ due to the scaling of the intermediate part of the left cut uncertainty (Eq. (68)).

## A.2 Illustration of the IAM's inefficiency if a CDD amplitude zero is near a resonance pole

### A.2.1 General discussion

To illustrate how to proceed to overcome the CDD difficulty in the IAM, it is enough to consider the following simple toy model of an unspecified partial wave [9] that features both a resonance and a CDD pole,

$$t(s) = \frac{1}{\frac{\mathfrak{f}^4}{s(s-M_0^2)} - i\sigma} \ , \tag{82}$$

with $\mathfrak{f}$ being an energy scale. We notice two zeroes of $t(s)$, the Adler zero at $s = 0$ and the CDD pole at $s = M_0^2$. In the second Riemann sheet the equation for the resonance pole is

$$\mathfrak{f}^4 - is(s - M_0^2)\sigma(s) = 0 \ , \tag{83}$$

with $\text{Im}[\sigma(s - i\epsilon)] < 0$. To keep the discussion simple, let us take the chiral limit $m_\pi \to 0$ to calculate the pole positions. For $m_\pi \to 0$, the analytical extrapolation of $\sigma(s)$ to $\text{Im}(s) < 0$ is $+1$ in the second and $-1$ in the first Riemann sheet, respectively. As a result, the secular equation to be solved for $\text{Im}(s_R) < 0$ is $\mathfrak{f}^4/s_R(s_R - M_0^2) - i = 0$, with the solution

$$s_R = \frac{M_0^2}{2}\left(1 + \sqrt{1 - i\frac{4\mathfrak{f}^4}{M_0^4}}\right) \ . \tag{84}$$

Let us examine what would the IAM predict for such amplitude. First, let us construct the chiral expansion (in powers of $s$) of $t(s)$ in Eq. (82), which is all that would be accessible from a low-energy measurement,

$$t(s) = -\frac{sM_0^2}{\mathfrak{f}^4} + \frac{s^2}{\mathfrak{f}^4}\left(1 + i\frac{M_0^4}{\mathfrak{f}^4}\sigma(s)\right) + \mathcal{O}(s^3) \ . \tag{85}$$

To more clearly express this expansion in notation familiar in ChPT and HEFT, we rewrite

$$\frac{M_0^2}{\mathfrak{f}^4} \equiv \frac{\lambda}{v^2} \implies v^2 = \frac{\lambda\mathfrak{f}^4}{M_0^2} \ , \tag{86}$$

where $\lambda$ is a positive numerical coefficient,[10] e.g. for the $P$-wave $\pi\pi$ scattering $\lambda = 1/6$, while $v$ is equivalent to $f_\pi$ [10]. Then, the expansion in Eq. (85) becomes

$$t(s) = -\frac{\lambda s}{v^2} + \frac{\lambda s^2}{M_0^2 v^2} + i\sigma(s)\left(\frac{\lambda s}{v^2}\right)^2 + \mathcal{O}(s^3) \ . \tag{87}$$

It is clear that the scale driving the chiral expansion is $M_0^2$. For instance, the NLO counterterm that can be read from Eq. (87) scales as $v^2/M_0^2$, of typical size in any chiral expansion. In terms of these parameters the pole position $s_R$ in Eq. (84) reads

$$s_R = \frac{M_0^2}{2}\left(1 + \sqrt{1 - i\frac{4v^2}{\lambda M_0^2}}\right) \ . \tag{88}$$

---

[9]The model is obviously inspired by $K$-matrix considerations, so it has theoretical deficiencies that are of no concern for this discussion, such as inadequate analytical behavior on the first Riemann sheet -poor implementation of causality- with a second pole, that fortunately is far from the physical region of interest.

[10]The factor $M_0^2$ relating $\mathfrak{f}^4$ to $v^2$ appears because the $t(s)$ in Eq. (82) can be considered a $K$-matrix unitarization of the chiral $s$-channel exchange of a resonance with constant propagator [10, 82].

The IAM construction would then extrapolate this expansion to higher physical $s$ taking the form

$$t_{IAM}(s) = -\left(\frac{\mathfrak{f}^4}{sM_0^2}\left(1+\frac{s}{M_0^2}\right)+i\sigma\right)^{-1} = -\left(\frac{v^2}{\lambda s}\left(1+\frac{s}{M_0^2}\right)+i\sigma\right)^{-1}. \tag{89}$$

The Adler zero at $s=0$ is recognizable, but the second zero, the CDD pole at $s=M_0^2$, has not been recovered by the IAM based on the expansion of Eq. (85):

Compare this to the analogous form that Eq. (82) takes in terms of $v$,

$$t(s) = -\left(\frac{v^2}{\lambda s}\left(\frac{M_0^2}{M_0^2-s}\right)+i\sigma\right)^{-1}. \tag{90}$$

Failure to reproduce the CDD pole by the standard IAM comes along with the lack of the original pole in the second Riemann sheet, suffering instead from one in the first Riemann sheet at

$$s_R = -\frac{M_0^2}{1+iM_0^4/\mathfrak{f}^4} = -\frac{M_0^2}{1+i\lambda M_0^2/v^2}. \tag{91}$$

The criterion of Eq. (17) to identify a resonance near the real axis,

$$t_0(s_R) - \operatorname{Re}t_1(s_R) = -\frac{M_0^2 s_R}{\mathfrak{f}^4} - \frac{s_R^2}{\mathfrak{f}^4} = 0, \tag{92}$$

whose solutions are 0 and $-M_0^2$, does not yield any sensible result with $\operatorname{Re}(s_R)>0$ either. It appears clear to us [42] that this difference between the actual wanted amplitude in Eq. (82) and the IAM is due to the near proximity of the CDD zero of the amplitude to the resonance pole, which is then masked to the IAM.

### A.2.2 Numerical implementation of the method in subsection. 4.1.3

To illustrate the procedure taking care of the CDD pole, let us look at a computation for the $I, J = 1, 1$ vector isovector channel with HEFT as deployed in [22]. The LO constants are chosen as $a = 0.9$, $b = a^2$, and the NLO ones as $a_5(\mu = 3\,\text{TeV}) = -1.75\times10^{-4}$, $a_4(\mu = 3\,\text{TeV}) = -1.5\times10^{-4}$ (with all others set to zero). This set yields a $\rho$-like resonance around 3-4 TeV as shown in figure 8.

For those values of the low-energy constants, there is no CDD pole so the standard IAM suffices [11]. If we increase the absolute value of $a_4(\mu = 3\,\text{TeV})$ to $-4.5\times10^{-4}$, a CDD pole appears satisfying Eq. (26), as shown in figure 9. We then calculate the conventional IAM of Eq. (12) and the CDD-modified IAM of Eq. (29), and plot it in Figure 10.

The difference between the IAM and the CDD-IAM is clear: the second naturally reproduces the CDD pole (zero of the amplitude), while the first fails to do so and starts taking off towards a resonance such as that of figure 8, just at higher mass (whether and when we would trust the method at that high energy is another discussion about predictivity reach not relevant for this point of the CDD zero; once this first structure is well behind, the CDD-IAM should not be used.).

To clarify what happened to the resonance of figure 8 upon changing the low-energy constant so a CDD pole of the inverse amplitude appears in this channel, we need to pursue the computation of the pole in the second Riemann sheet.

---

[11]In figure 12 below we show a calculation of the pole position of the resonance in the second Riemann sheet. For the same parameter set, the real part of the pole position $s_P$ differs from the nominal mass on the real axis by 5%. Therefore, at this width, the uncertainty of computing on the real $s$ axis and not in the complex plane is subleading to the uncertainty introduced by the left cut (at 17%). In any case, a more detailed analysis can beat that 5% by propagating errors taking into account the complex-number nature of the resonance position.

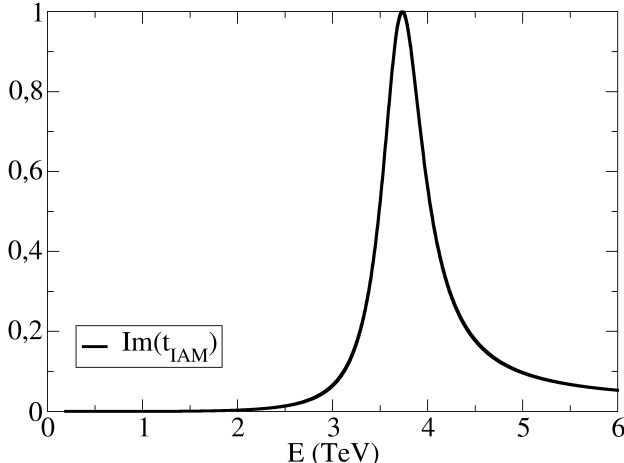

Figure 8: A simple $\rho$-like resonance in $\omega\omega \simeq W_L W_L$ scattering in the HEFT formulation.

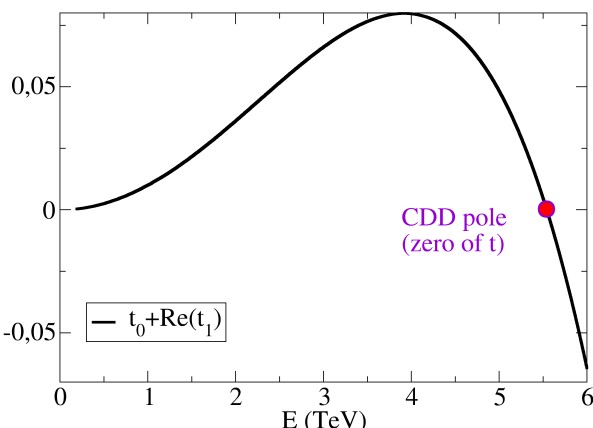

Figure 9: If the value of $a_4$ is made more negative than in figure 8, a CDD pole (zero of the partial wave amplitude) appears, as shown by $t_0 + \text{Re}(t_1) = 0$.

Figure 11 shows the amplitude in the first (top plots) and second (bottom plot) Riemann sheets for $a_4(\mu = 3\text{TeV}) = -2.5 \times 10^{-4}$. A branching point and cut along the real $s$ axis is clearly visible in the first one, whereas the second shows an amplitude growing towards a pole for negative imaginary $s$ as expected.

Once this has been visually checked, we can follow the movement of the resonance pole in the complex plane in the second Riemann sheet, shown in figure 12. The computed data has employed the bare IAM: because the resonance is broad and far from the axis for a large swath of the parameter space described in the figure, the method is able to capture it in spite of not reproducing the shape along the physical real axis.

If we instead employ the modified IAM from Eq. (29), with parameters $a = 0.9$, $a_4 = -4.5 \times 10^{-4}$, $a_5 = -1.5 \times 10^{-4}$, the pole sits, approximately, at $s = (15.7, -96.1)\text{GeV}^2$ on the second Riemann sheet, corresponding to $M \simeq 4$ GeV, $\Gamma \simeq 24$ MeV, that is, extremely

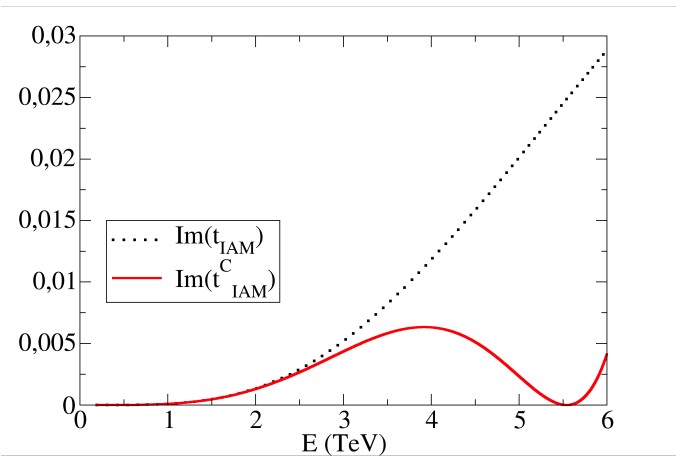

Figure 10: The conventional IAM (dotted) applied to the amplitude $t$ fails to reproduce the zero associated to a CDD pole. However, applying the method to the amplitude $t/(s - s_C)$, leading to the small modification of Eq. (29), correctly reproduces the CDD zero in the amplitude (solid line).

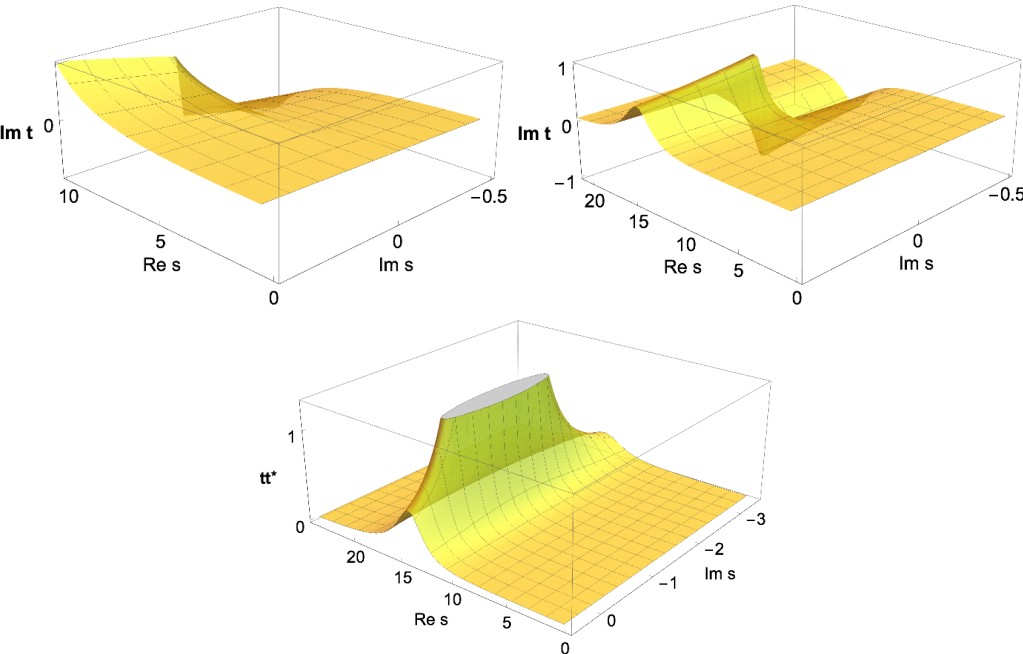

Figure 11: The resonance of figure 7 using the IAM extended to the complex $s$ plane, with the parameters $a = 0.95$, $a_4 = -2.5 \cdot 10^{-4}$ and $a_5 = -1.75 \cdot 10^{-4}$. The two top plots show the first Riemann sheet, cut along the real axis $\mathrm{Im}(s) = 0$. The left one is limited to low and moderate $s$, and only the cut is visible; the right one extends to the resonance energy that, although it does not leave a pole in this first sheet, it is seen to saturate unitarity ($\mathrm{Im}(t) \simeq 1$ for a certain real $s$ (the scale does not allow to visualize the cut). The plot in the bottom line is the extension to the second Riemann sheet and clearly features a pole for negative $\mathrm{Im}(s) < 0$.

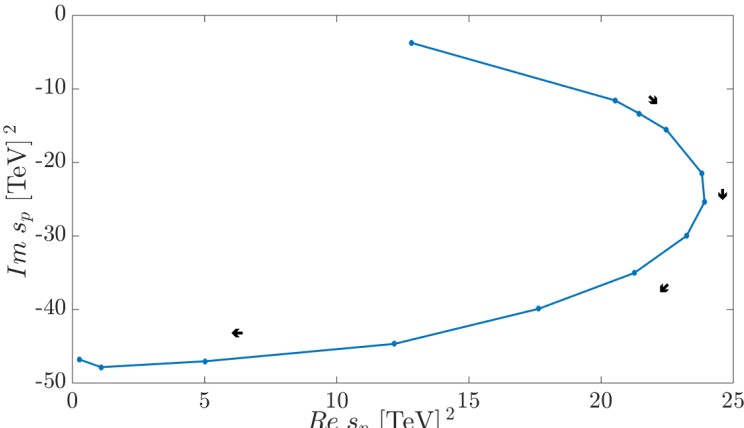

Figure 12: Motion of a resonance pole in the complex plane, for $a_4 \in [-3.56, -1.5] \cdot 10^{-4}$, $a_5 = -1.75 \cdot 10^{-4}$ and $a = b^{1/2} = 0.9$ for the basic IAM. The arrows indicate the flow of the pole position with increasing absolute value for $a_4$.

deep in the complex $s$-plane, in agreement with the basic IAM. The difference is sufficiently substantial to much prefer the use of the modified method also for the broadest resonances, due to the effect of the missed zero in the basic IAM.

### A.3 Left cut partial wave amplitude: a new characterization

In this appendix we provide a (possibly new) characterization of the partial wave over the left cut, which might be useful for uncertainty estimates, but we have not yet fully exploited it and have decided to relegate it off the main text.

The partial wave over the left cut, with $s \leq 0$, is

$$t_{IJ}(s) = \frac{1}{32\pi\eta} \int_{-1}^{+1} dx \, P_J(x) \, T^I(s, t(x), u(x)), \tag{93}$$

where $t(x) = (2m^2 - s/2)(1 - x)$ and $u(x) = (2m^2 - s/2)(1 + x)$. Following Mandelstam, we assume that there is a unique analytic amplitude $T(s, t, u)$ such that

$$T(s, t, u) = \begin{cases} T_s(s, t, u), & \text{for } s \geq 4m^2,\, t \leq 0,\, u \leq 0\,, \\ T_t(t, s, u), & \text{for } t \geq 4m^2,\, s \leq 0,\, u \leq 0\,, \\ T_u(u, t, s), & \text{for } u \geq 4m^2,\, t \leq 0,\, s \leq 0\,, \end{cases}$$

with these three physical regions of different channels shaded in Fig. 13 (misty rose color online) .

Notice that the partial-wave interval of integration in Eq. (93) corresponds to a line crossing the patterned region (purplish online) of the Mandelstam plane shown in Fig. 13, with endpoints fixed at the physical $u$- and $t$-channel borderlines (velvet online).

Hence we see that, for the integral which defines the partial wave over the left cut, the endpoint values of the integrand are amenable to treatment from experimental data. These values correspond to the physical amplitude below its right cut in the $u$- and $t$-channels respectively,

$$\begin{cases} T(s, x = +1) = T(s, 0, 4m^2 - s) = T_u(4m^2 - s, 0, s)\,, \\ T(s, x = -1) = T(s, 4m^2 - s, 0) = T_t(4m^2 - s, s, 0)\,, \end{cases}$$

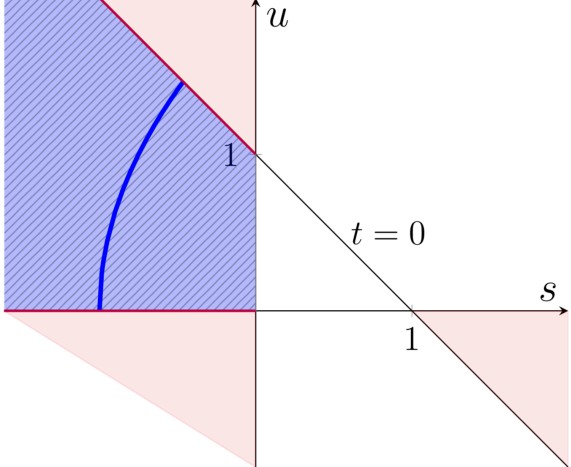

Figure 13: Mandelstam plane for two identical particles (in units of $4m^2$, $t = 1-s-u$). The three $2 \to 2$ physical regions (misty rose-coloured online) extend outside the triangle vertices. In the center-left (patterned, purplish online), we show the un-physical region where the integrand in Eq. (93) is evaluated. The arch (thick, blue online) represents the contour of integration for $x \in [-1, 1]$; the argument of the partial wave on the left cut, $s$, is negative. Note that along the slanted thick seg-ments (velvet online) the kinematics correspond to the physical $u$- and $t$-channels, so the amplitude can be evaluated from data at the end points of the arch.

since the left cut integrand of the $s$-channel (on the left-hand side) is evaluated at $s + i\epsilon$ with $s \leq 0$ and therefore the combination $4m^2 - s - i\epsilon$ lies below the right cut of the $u$ or $t$ channels on the RHS.

The partial-wave of Eq. (93) is a linear combination of the auxiliary functions (omitting isospin indices),

$$\psi_J(s) \equiv \int_{-1}^{+1} dx\, x^J\, T(s, t(x), u(x))\,, \tag{94}$$

which we will integrate by parts *ad infinitum*.

These infinitely many integrations by parts can be carried out due to Cauchy's inequalities, since they guarantee that the $n$-th derivative respect to $x$ of the amplitude is bounded in modulus by a quantity of order $n! \times |\sup(T)|$ in the region of the complex $x$-plane where $T$ is analytic. Successively integrating ($n$ times) the monomial $x^J$ yields $x^{n+J} J!/(n+J)!$. Observe then that the denominator controls the $n$-th derivative of the amplitude and the integral of the monomial $x^{J+n}$ tends to zero when $n \to \infty$. Hence,

$$\psi_J(s) = \sum_{n=0}^{\infty} \frac{(-1)^n J!}{(n+1+J)!} \left( \frac{\partial^n}{\partial x^n} T(s, x) \Big|_{x=+1} + (-1)^{n+J} \frac{\partial^n}{\partial x^n} T(s, x) \Big|_{x=-1} \right). \tag{95}$$

Note that

$$\frac{\partial^n}{\partial x^n} T(s, x) \Big|_{x=+1} = (2m^2 - s/2)^n \frac{\partial^n}{\partial u^n} T_u(u, 0, s) \Big|_{u=4m^2-s}\,, \tag{96}$$

with $u = 4m^2 - s$, and

$$\frac{\partial^n}{\partial x^n} T(s, x) \Big|_{x=-1} = (s/2 - 2m^2)^n \frac{\partial^n}{\partial t^n} T_t(t, s, 0) \Big|_{t=4m^2-s}\,, \tag{97}$$

with $t = 4m^2 - s$. So that these derivatives are to be taken from the amplitudes in both $u$- and $t$- physical channels. In the case of purely elastic scattering the only branch points are at each

channel's two pion threshold. Hence, we exclude $s = 0$ where the branch points of $u$- and $t$-channels sit for $x = \pm 1$ respectively from this treatment. The derivatives can be taken since the functions are evaluated in the first Riemann sheet, where the amplitude is analytic. Here we are also assuming that the supremum of $T$, $\sup T$, exists in a neighbourhood of the cuts.

To pass from the monomials $x_J$ to the Legendre polynomials, we represent them as

$$P_J(x) = 2^J \sum_{k=0}^{J} \binom{J}{k} \binom{\frac{J+k-1}{2}}{J}_g x^k, \tag{98}$$

where the generalized binomial coefficient is

$$\binom{\alpha}{k}_g = \frac{\alpha(\alpha-1)(\alpha-2)...(\alpha-k+1)}{k!}, \tag{99}$$

to express the partial wave over the left cut ($s < 0$) as

$$t_J(s) = \frac{2^J}{32\pi\eta} \sum_{k=0}^{J} \binom{J}{k} \binom{\frac{J+k-1}{2}}{J}_g \sum_{n=0}^{\infty} \frac{(-1)^n k!}{(n+1+k)!} \times$$
$$\times \left( \frac{\partial^n}{\partial x^n} T(s,x) \Big|_{x=+1} + (-1)^{n+J} \frac{\partial^n}{\partial x^n} T(s,x) \Big|_{x=-1} \right). \tag{100}$$

The advantage of this expression is that the partial wave over an unphysical line (the left cut) is expressed in terms of quantities evaluated a two physical points corresponding to the $u$- and $t$-channel right cuts respectively. Studies such as [83] review dispersion relations for the derivatives of the forward amplitude appearing in Eq. (100) and could be useful for future studies. Now, we could be tempted to use the unitarity relations for the physical amplitudes,

$$-2\operatorname{Im} T_u(4m^2-s, 0, s) = T_u(4m^2-s, 0, s) T_u^*(4m^2-s, 0, s) \tag{101}$$

and

$$-2\operatorname{Im} T_t(4m^2-s, s, 0) = T_t(4m^2-s, s, 0) T_t^*(4m^2-s, s, 0), \tag{102}$$

to relate $\operatorname{Im} t_J(s)$ to $|t_J(s)|^2$ over the left cut (remember that $\operatorname{Im} G = -(t_0^2 \operatorname{Im} t)/|t|^2$), and use derivatives of these expressions to address Eq. (100). We may pursue this line of thought in future work.

## A.4 Uncertainty on the intermediate left cut if $\operatorname{Im}(G + t_1)$ does not change sign

Under the assumption that $\operatorname{Im} G + \operatorname{Im} t_1$ does not change sign over the left cut (i.e. $\operatorname{Im} G$ stays always below or always above $-\operatorname{Im} t_1$), it is possible to put an overall bound to the left cut integral of Eq. (15) different from our effort in the main text. To do so note that, in the physical region $s > 4m^2$,

$$\left| \int_{-\infty}^{0} dz \frac{\operatorname{Im} G + \operatorname{Im} t_1}{z^3(z-s)} \right| \leq \left| \int_{-\infty}^{0} dz \frac{\operatorname{Im} G + \operatorname{Im} t_1}{z^3(s_0 + |z|)} \right|, \tag{103}$$

for an $s_0 = 4m^2 - \epsilon$ below threshold in the real axis (because $|z - s| \geq |z| + s_0$ while $z < 0$). We choose $s_0$ as needed to avoid it to coincide with an Adler zero (that causes a divergence of the inverse function $G$). Now we make use of a dispersion relation similar to Eq. (10) but for the function $G(s) + t_1(s)$ evaluated at $s_0$. After neglecting the Adler zeroes of $G$ (the uncertainty introduced has already been evaluated in subsection 4.1), we find

$$\left| \int_{-\infty}^{0} dz \frac{\operatorname{Im} G(z) + \operatorname{Im} t_1}{z^3(z-s)} \right| \leq$$
$$\leq \frac{\pi}{s_0^3} \left| \left[ G(s_0) - G(0) - G'(0)s_0 - \frac{1}{2}G''(0)s_0^2 \right] + \left[ t_1(s_0) - t_1(0) - t_1'(0)s_0 - \frac{1}{2}t_1''(0)s_0^2 \right] \right|, \tag{104}$$

due to $RC(G + t_1) = 0$ by virtue of Eq. (5).

To treat the first square bracket of the RHS of Eq. (104), we will choose $s_0$ between 0 and the first Adler zero (below the first threshold). In the interval $[0, s_0]$ we can assure that $G$, a real function, is continuous and with continuous derivative. In this way we have

$$\left[ G(s_0) - G(0) - G'(0)s_0 - \frac{1}{2}G''(0)s_0^2 \right] = R_2(s_0) \,, \tag{105}$$

where the Taylor remainder $R_2(s_0)$ equals

$$R_2(s_0) = \frac{1}{2} \int_0^{s_0} ds \, G'''(s)(s_0 - s)^2 \,. \tag{106}$$

Now, since we are deep in the Chiral Perturbation Theory region (i.e. very low $s$), we have (ignoring the slow-variation logarithms of perturbation theory against the powers $G = g_0 + g_1 s + g_2 s^2 + g_3 s^3 + ...$), $G'''(s) = 3! \, g_3 \big(1 + \mathcal{O}(s/f_\pi^2)\big)$ to obtain

$$\frac{\pi}{s_0^3} R_2(s_0) = -\pi g_3 \big(1 + \mathcal{O}\big(\frac{s_0}{(4\pi f_\pi)^2}\big)\big) \,. \tag{107}$$

From this equation we see that the Taylor remainder takes a very small uncertainty $\mathcal{O}(s/f_\pi^2)$, which is a correction of the correction and duly neglected.

Remembering the $\mathcal{O}(p^6)$ treatment in section 4.4, from Eq. (49) we have that

$$3! \, g_3 = \Big( \frac{t_1^2}{t_0} - t_2 \Big)'''(0) \,. \tag{108}$$

The estimation (neglecting logarithmic contributions) of the imprint of a high energy resonance in the third order constant $g_3$, can be obtained from [55] (equation (A4) in the reference). For the $IJ = 11$ channel we have that,

$$g_3 = \frac{40{l_1}^2 - 40l_1 l_2 + 10{l_2}^2 - 9(r_5 + r_6)}{960\pi f_\pi^6} \,. \tag{109}$$

All the constants needed in (109) are presented in Table 2.

Because in neglecting the logarithms we are employing a polynomial approximation for $G$ and $t$ in the low-energy interval of interest (that excludes any cuts or imaginary parts of the amplitudes),

$$\frac{\pi}{s_0^3} \Big[ t_1(s_0) - t_1(0) - t_1'(0)s_0 - \frac{1}{2}t_1''(0)s_0^2 \Big] \simeq 0 \,. \tag{110}$$

Hence, the overall bound for the uncertainty introduced by approximating the left cut is, using (15),

$$|\Delta(s)G(s)| \le s^3 |g_3| \simeq 6.7 \cdot 10^{-2} \,, \tag{111}$$

which is of the same order as the computed in the text. However the hypothesis assumed here remains open. A check of a computation of Im $G$ over the left cut with the Roy-equations shows that there is no zero of the amplitude (though this can be small) so there is no pole of $G$ in this 11 channel [2, 43] down to about $-1\text{GeV}^2$. Since this may be outside the zone of applicability of the method because exceeding the Lehman ellipse not allowing the partial wave expansion for that very negative $s$, the calculation needs to be taken with much caution.

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
