# Peer review of "Systematizing and addressing theory uncertainties of unitarization with the Inverse Amplitude Method"

_SciPost Physics, doi:SciPost Phys. 11, 020 (2021)_

## Round 2 · Referee Report · Jose Pelaez · 2021-7-2

Strengths

1) The paper presents a breakthrough on a previously-identified and long-standing research stumbling block, namely, the quantification of systematic uncertainties in unitarization of scattering in effective theories.

2) The paper opens a new pathway in an existing direction, with clear potential for multipronged follow-up work, since now the estimates of systematic uncertainties can be addresses in the different fields where unitarization of EFTs is used. It also sets the standard for the assessment of uncertainties in other unitarization methods beyond the IAM (N/D, Chiral Unitary approach, K-matrix...) Maybe it can even lead the way for other systematic uncertainty assessments beyond scattering.

3) The paper definitely provides a novel and synergetic link between different, research areas, since the assessment checks using Hadron Physics data can be translated for to electroweak symmetry breaking studies and EFTs on BSM physics.

Weaknesses

I see none

Report

As explained in the "strengths" section above, the paper meets three of the expectations required to be published in SciPost. It also fulfills all the general acceptance criteria on the clearness of its composition, detailed abstract, enough detail to allow its results to be reproducible by qualified experts, an appropriate list of citations to relevant work on the topic, as well as a clear conclusion summarizing the results.

Requested changes

No changes requested.
The authors have answered appropriately all the cavetas and suggestions raised in my previous report.

---

## Round 2 · Author Response

We thank the referee for the careful assessment of the manuscript and for detailing some strengths; we, on our side, have attempted to alleviate his/her perceived weaknesses as described below.
As the referee indicates this opens new avenues for application to other resonances of interest in hadron physics or in the electroweak sector, which is left for future work for this or other research teams. For each such specific application one could also rather straightforwardly consider the systematic uncertainties stemming from the errors affecting the EFT counterterms on top of the ones considered here. We now provide an example in the manuscript in section 5 as detailed below.

---

## Round 2 · List of Changes

REPORT:
As for the SciPost expectations criteria, to my view the manuscript presents a breakthrough on a previously-identified and long-standing research stumbling block, which is the estimation of uncertainties in unitarization methods and it opens a new pathway in similar studies for other unitarization methods applied to different effective theories.

Concerning the general acceptance criteria for SciPost, I think they will be easily met after the authors address properly the comments that I detail next:

a) The authors make a valuable effort to ascertain the systematic uncertainties in the IAM and one of their main motivations is the prediction of New Physics states or resonances that may lie out of reach of LHC but whose non-resonant effect may be observed at low energies. Here I see several issues that should be commented or addressed by the authors.

a.1) Their generic estimates are made in comparison with a rho-like scenario. This imposes a size for the low-energy constants (LECS), given by Resonance Saturation and the ratio f/Mrho. However, if the LECS were much smaller because the new states lied much higher compared to v (246 GeV), the equivalent ratio v/Mnew would be much smaller and I suspect that all the relative uncertainties calculated here would be much larger. It should be easy for the authors to estimate how high the resonances should be so that the IAM becomes useless due to its systematic uncertainties. Although the authors explicitly say that their estimates are obtained in a rho-like scenario, it is relevant to provide at least another example with higher masses to illustrate the scaling of the uncertainties. (There is also the issue that the predicted resonance could be as wide as massive so that it may be hard to identify as a resonance)

RESPONSE: We are presenting the method to systematize the uncertainties, not the uncertainty applicable to all relevant cases. As the referee requests, we have commented on the scaling of the uncertainties leading to failure of the method in a new subsection 5.1, as the mass of the new physics is pushed higher. A discussion on the case of the $\sigma$ resonance has been also included, comprising the fifth paragraph before Sec.5.2. At 40+ pages, we however think that entertaining another complete example with different J^PC in a different channel would put the article out of bounds and have abstained from doing so beyond sporadic comments. As for a resonance that has comparable width and mass would indeed be an experimental encumbrance, but the method has no trouble with it as shown in the new figure 12 that follows a resonant pole deeply in the complex plane.

REPORT:
a.2) The work deals with longitudinal gauge boson and/or Higgs scattering. It is however possible that some of these states may be better seen in other processes (in the case of a rho-like state, the resonance may be much more easily seen through vector form factors starting from gluon fusion without the need for "initial" longitudinal gauge bosons). In such case, non-resonant behavior may not have been clearly identified in scattering but a resonance may have been seen in another process related through final state interactions. The specific scenarios to which the authors refer, where the IAM would suggest new resonant physics should be commented with more detail. In particular, what if the resonance is not seen in scattering but in other processes? What to do with the IAM then?

RESPONSE: The IAM has been applied to obtain such form factors and more generally the footprint of a resonance of the Goldstone/Higgs sector in other channels. We have added a new subsection 5.3 with an explanation of the procedure. In brief, the uncertainty in the phase of a production amplitude and the position of a resonance pole therein, is identical to the corresponding identity in the elastic amplitude, due to Watson's final state theorem.

REPORT:
a.3) There is the issue of the uncertainties in the LECs as well. If the resonance lies out of reach of the LHC and only its "low-energy tail" is seen as a non-resonant effect, the corresponding LECS will have, I am afraid, a rather large uncertainty. This actually happened in the first applications of the IAM to the rho, when the uncertainties in the LECS were so big that their propagation to the rho pole using the IAM covered very well the correct description of data, around the resonance peak, dwarfing the systematic IAM uncertainties. The question is, then, whether the uncertainty in the measured (non-resonant) deviations from the SM will dominate over the IAM uncertainty. This should be commented and, of course, that will change with the mass of the resonance (see a.1 above)

RESPONSE: The referee brings up a different type of uncertainty which is induced by experimental data,
not by the theory method itself. We believe that we have made very clear that we are addressing the systematic uncertainties of the method itself, not the statistical uncertainties that require an entirely
separate treatment (but this is more standard work, propagating them through the entire chain of formulae).
Still, we understand that the referee is curious about the relative size of uncertainties. Therefore, we have dedicated subsection 5.2 to the exercise of varying the LECs to estimate how much do they need to vary for their uncertainty to equate the systematic theory one. It is seen (at NLO)
that, until their eventual separation from the Standard Model is known at the level of 30 percent, the uncertainty considered in this article is subdominant, as the referee correctly points out.

REPORT:
b) CDD poles are not so much, or not only, a source of uncertainty as an impediment to apply the naive IAM, questioning the validity of the assumptions needed to derive it. The examples about CDD poles in Appendix A.2 need some clarification and some more physical insight of why this is so. I understand the need for a simple model, but the fact that f is both the resonance pole mass (Eq.80), the scale for the chiral expansion (Eq.81 in s/f^2) and of the same order as the new dynamical scale M_0 makes the discussion somewhat confusing. In particular:

b.1) The authors state that Eq.81 is a well-behaved chiral expansion. I agree that it is a low-energy expansion, but why they claim is chiral? How would one arrive to that expression from a chiral Lagrangian? Surprisingly the LO seems to depend on more parameters than just f. How would M_0 appear there?. Why f is to be identified with a chiral scale? From what it seems, it is just the resonance mass, and if it is also the chiral scale then the resonance is very light compared to the hadronic cases, (almost by an order of magnitude compared to the rho). Actually, the fact that t0 and Re t1 cancel before s= f^2 is not just evidence for the breakup of the series? Certainly we can construct an amplitude like that, but zeros appear for some dynamical reason, so the authors are setting the new dynamical scale at M_0 (below f), similar to f but much below the "chiral scale" 4πf which is the order of magnitude where the relevant degrees of freedom of the theory are integrated out to buid the EFT. But nowhere is another dynamical scale M_0. I am not so sure this is a chiral expansion in the usual sense, the authors should definitely comment about this.

RESPONSE: We agree that the way Eq.(81) was originally written in terms of $f$ could drive to confusion. We have improved upon the clarity of the presentation of the model in several aspects that we think satisfy the questions of the referee. In particular, Eq.(86) rewrites the expansion of Eq.(84) in a way that comforts the standard manner in which chiral series are presented, this is why we call it chiral. Because of the relation in Eq.(85) it is clear that the expansion is in powers of $s/M_0^2$. The original energy scale $f$ is now denoted $\mathfrak{f}$, so as to avoid identifying it directly with the chiral order parameter, that corresponds to $v$.

As indicated in the new footnote 10, $t(s)$ in Eq.(81) "The factor $M_0^2$ appears in the relation between $\mathfrak{f}^4$ and $v^2$ because $t(s)$ in Eq.~(81) can be considered as the K-matrix unitarization of the chiral $s$-channel exchange of a resonance in which its propagator has been frozen to a constant [8, 71]." For instance the referee can consult Eq.(49) of Ref.[9], or the first two terms on the right-hand side of Eq.(81) in Ref.[76].

The resonance pole position is given in Eq.(87) within the new notation. It is driven by $M_0^2$. For the case of the $\pi\pi$ $P$-wave scattering $1/\lambda=6$, which is rather large so that the imaginary part in the square root also makes an impact in $s_R$. Therefore, the resonance is not light compared to the hadronic scale, which determines $M_0$. In QCD the unitarity scale $4\pi f_\pi$ and the hadronic scale given by the resonance masses both approximately coincide, being around 1~GeV~$M_0$.

REPORT:
b.2) The authors are assuming f and M_0 to be close, with M_0 somewhat smaller... how justified is then the expansion in Eq.80?

RESPONSE: After the rewriting mentioned in b.1) this is now clearly not the case. The referee can consult Eqs.(86) and (87). $v$ is the chiral scale analogous to $v$ in HEFT or $f_\pi$ in ChPT, while $M_0$ is the hard scale in the expansion.

REPORT:
b.3) How well the expansion in 81 converges compared to the usual ChPT case?. In ChPT the LECS are of order 10-2, or, in other words, we expect a suppression with powers of 4πf (as the authors remark in several places in their work). Where is that suppression in Eq.81?, it seems that the suppressing 4π factors are gone. Actually, the presence of M_0/f factors is diminishing the LO wrt the NLO in the Hadron Physics case.

RESPONSE: This is now clearly answered by Eqs.(85) and (86). The NLO counterterm in the latter is $\lambda v^2/M_0^2$. By taking $\lambda=1/6$ as in the case of $P-wave$ $\pi\pi$ scattering, $v=f_\pi$ and with $M_0\simeq 1$~GeV we have that this counterterm is $1.4\times 10^{-3}$. If one consider $\lambda\simeq 1$ then it is $8\times 10^{-2}$, perfectly reasonable values for a NLO ChPT counterterm.

REPORT:
b.4) Is really the IAM to be blamed for not reproducing well the resonance? Or is it just that the NLO low-energy expansion is the one not able to reproduce a zero at s=M^2_0 (the real parts do cancel, but the imaginary part does not and its coefficient is order one, if I understand correctly). I think the the blame can also be put on the effective theory, which is a very bad approximation to the zero at M_0. Of course, if one removes the CDD pole, the rest of the amplitude has a convergent expansion on which unitarization can provide meaningful results. But it seems more a problem of the expansion itself than the unitarization. This should also be discussed an commented.

RESPONSE: As we have shown in the previous points there is nothing pathological affecting the NLO chiral expansion in this toy-model. Because of unitarity, a zero of $t(s)$ can occur when the phase shift of the elastic partial-wave amplitude is $0\,\text{mod}\,\pi$. Then the imaginary part is zero, as the real part. A zero could be mimicked by a perturbative low-energy expansion of the interactions among pions or longitudinal components of the $W$ and $Z$, while a resonance phenomenon cannot. This point is further extended when replying point c) of the referee's report.

REPORT:
b.5) Another way of seeing the previous comments is that, naively, the IAM reconstructs a resonance from the information on its low-energy tail. This information can be encoded in the LECS. Indeed, for Hadron Physics, it is known that the value of the combination of l_1 and l_2 parameters in the rho channel are dominated by the rho contribution itself (This is known as Resonance Saturation). The smallness of the combination that appears in the scalar channel and the enhanced factor of the loops there make the sigma a very different kind of resonance, but if the LECS had a different value it would look relatively similar to the rho again. The authors should discuss whether by placing the CDD pole "right before" the resonance (Since M_0 is slightly less than f), the information about the resonance in the LECS becomes subdominant, which hinders its reconstruction with the IAM unless the additional and dominant dynamical information about the presence of the M_0 scale is provided and removed. Unfortunately, with the model used by the authors, where the resonance mass seems to be given by f, it is hard to disentangle in the EFT series what is the information about the resonance and what is the scale of the chiral breaking (assuming they explain why that is a chiral series, see b.1 above).

RESPONSE: In the new version we have removed the remark that $M_0\lesssim f$ because it would correspond to a very small $\lambda$, while $\lambda$ is expected to be a number of order 1. When giving $s_R$ in the previous version we introduced this expansion in powers of $(M_0/f)^4$ as an algebraic device to provide simpler algebraic expressions in this limit which, on the other hand, we agree with the referee that it is not a natural one when discussing unitarization of a low-energy chiral EFT. We think that the new Eq.(87) makes clear how the resonance mass-scale $M_0$ is providing the chiral-expansion scale while $v$ is the usual spontaneous chiral-symmetry breaking scale.

Within the toy-model in the manuscript it is manifestly clear that the counterterms are saturated by the resonance exchange, since one has a straight expansion in powers of $s/M_0^2$ without any cancellation. This is also the case for the model of Ref.[9] mentioned before in replying b.1).
There, in the channel of the $\rho(770)$, a CDD pole can be discussed and its position modulated by parameterizing the deviation from the KSFR relation with a parameter $g_v^2$. If $g_v^2=1$ then the KSFR relation is exact and the CDD pole moves to infinity. On the other hand, if $g_v^2<1$ and tends to smaller values then the CDD pole can appear $\lesssim 1$~GeV, and it can even be made as close as wished to the resonance mass $m_\rho$. E.g. the referee can consult Eq.(49) of Ref.[9] where the the deviation with respect to the KSFR relation is parameterized by $g_v^2$ and the discussion on the dependence of the CDD pole position with this parameter ($s_2$ in Eq.(50) of this reference).

REPORT:
b.6) The main example in the appendix, although simple, at first sight does not seem to fit the motivation of the authors for a measurement of a non-resonant behavior without seeing the resonance, since the resonance would be located roughly at f ≃ 250 GeV, and I guess it would be seen even better than the non-resonant effects. In the same appendix, the authors provide then another example with HEFT. But here the CDD pole appears {\it past} the resonance. So the mechanism looks rather different to the previous example where the zero seem to be "hiding" the resonance from threshold. So, the question is whether, and how, is it also cancelling the resonance saturation of the LECS. Certainly here the LO and NLO do not cancel at low energies, which does not seem to compromise the good convergence of the series.

RESPONSE: After the clarifications introduced in the new version of Appendix A.2 it is clear that the resonance mass is driven by $M_0$ which can be taken to be much larger than $v=250$~GeV, and then the toy-model is not at odds with our motivation alluded by the referee. By the same token the two parts in the Appendix A.2 do not contradict each other and in both cases there is no cancellation of the LO and NLO at low energies. The CDD pole always appears at a position of ${\cal O}(p^0)$.
As for the second example, we now show the position of the resonance in the complex plane, and it is seen to migrate quite far by the time the CDD pole appears. The bump before that CDD is not resonant, it simply reflects that a positive, smooth amplitude needs to be concave between two zeroes.

We do not see a problem with the saturation of the counterterms by resonance exchange, as explained in b.5) for the toy-model which exhibits this fact by construction, or the $s$-channel $\rho$ exchange in Ref.[9]. In our opinion it is not a matter of canceling the resonance contribution to the counterterms but simply the fact that a resonance exchange (at tree-level) gives rise to a rational function with a numerator and a denominator. The latter is driven by the resonance propagator while the former has zeros whose positions depend on the parameters characterizing the coupling of the resonance (KSFR relation for the $\rho$ case), together with some numerical factors in the LO and resonance contributions. As said, this can be controlled explicitly by the equation above for the $s$-channel exchange of the $\rho$ resonance through the parameter $g_v^2$. As the resonance becomes narrower, i.e. $g_v^2$ diminishes, then the CDD poles approaches the resonance mass and eventually they cancel each other for $g_v^2\to 0$, cf. Eq.~(50) of [9].

REPORT:
b.7) How well is the criterion of "narrow resonance", which the authors decided to apply in their systematic estimates, apply to the resonance in Fig.8?
It does not look as narrow as the rho-like in Figure 6. Although the CDD zero distorts its shape, it looks like its width is comparable to the mass. Has the use of narrow-resonance criteria an additional effect hindering the reconstruction of that resonance or the estimation of uncertainties in this case?

RESPONSE: The expanded appendix A.2 includes a new figure 12 with the position of this example pole in the second Riemann sheet, for the parameters of figure 8, and its motion as it sinks deeper along the imaginary axis when the parameters are changed (for example, to try to get a CDD pole). For the parameters of figure 8, the difference between the real part of the pole position and the mass along the real axis, as stated in the new footnote 11, is of order 5\% and a subleading uncertainty to that caused by the approximate left cut. In principle, this uncertainty source is reducible by repeating the error propagation while keeping s complex.

REPORT:
c) When discussing the presence of the CDD pole Eq.24, obtained at NLO, is often used as implying the presence of a zero in the full (all orders) partial wave. But, at first sight,it only implies the vanishing of the real part of the LO+NLO. I understand that if there is a CDD pole one would expect that relation to hold approximately somewhere near the real CDD pole position. But I do not see the reciprocal implication. The authors also talk about its "correct position". That is known if you know the model a priori, and is certainly correct for the example. But... is it "correct" in general or the position is only approximated up to a given order in the series ?(As it happens with the modified IAM and the Adler zero). The authors should elaborate more on these questions.

RESPONSE: We have added a new paragraph after Eq.(25) explaining that, while a perturbative calculation should not properly reproduce a resonance, which is why we have to use a unitarization method in the first place, the occurrence of the zero in the real part of the LO+NLO partial-wave amplitude should better correspond to an actual zero of the amplitude, a feature that a perturbative expansion could mimic much better.

Regarding the last sentence of the referee, we already included in the first version of the manuscript the sentence two lines below Eq.(26): "To do it, we noted that $s_C={\cal O}(p^0)$ (because $s_C$ is not an Adler zero, it is a large scale)." In the new version we have removed the parenthesis since the explanation included is relevant. That is, one evaluates $s_C$ from the amplitude given at the order in which the chiral series is known and treats it as ${\cal O}(p^0)$ for the expansion of $\text{Re}[1/t^C(s)]$, as indicated in the line just after Eq.(26).

REPORT:
d) In section 4.2. I guess σi(s) is the same as ϕ2 or proportional to it. Please, for clarity, write the relation between the two quantities. I understand then the need for a dimensional factor to make a meaningful comparison with ϕ4. But, why the authors decide to multiply ϕ2 by powers of fπ?. They are making an estimate based on phase space, i.e. kinematics, whereas fπ is a dynamical feature. I understand that using mπ, which is a kinematic quantity would give problems in the chiral limit, and fπ is the only dimensionful quantity around. However,
the issue here is how much the four-pion state yields an inelasticity in two-pion scattering, i.e, their contribution to Im t(s). Why then fπ and not 4π fπ?. After all, the 4 pions appear in a two-loop diagram when they contribute to the pion-pion scattering amplitude. The authors even consider later 4π fπ as the suppression scale for other factors. Please explain why this difference.

RESPONSE: We see the referee's point that this section needed further and clearer explanation. To our understanding, the right comparison between phase spaces is the one already used. This can be understood by examining the unitarity relation for the amplitude above the four particle threshold (a good way to quickly
see it is cutting the loop-diagrams as in the new figure 2).

When we consider each of the 2- or 4-body phase spaces, we have assigned 2pi factors according to the normalization of one-pion states. For each pion in an intermediate state there is a factor 1/(2pi)^4. However, when calculating the imaginary part of the amplitude, each cut line will produce an extra 2pi (together with the $f_pi^-1$). This leaves the typical (2pi)^{-3} normalization factor for each pion. Hence, we must only use f_pi^4 to compare the two-body and four-body phase spaces.

REPORT:
e) There is an important caveat about the comment section 4.6 on crossing symmetry violation. The Roskies relations are certainly appropriate to quantify such violation. However, in the works of Cavalcante and Sa Borges, the IAM is used {\it once it has been fit to data}. As a consequence, what those authors have actually tested with their approach is how well {\it those data in the fit} satisfy crossing symmetry, not the IAM violation of crossing. In other words, any other curve/method/model/freehand-drawing fitting those data would yield the same values for the Roskies relations. Thus, that approach, {\it as applied by Cavalcante and Sa Borges}, does not test the IAM, nor any model, just the data. The authors should reconsider the writing of this subsection in view of this comment.

RESPONSE: We agree that there is a dependence on the fit to experimental data induced into the amount of Crossing Symmetry violation present in the IAM's amplitudes. This applies to the work [67] https://arxiv.org/pdf/hep-ph/0110392.pdf
where it is unclear that providing another parametrization would improve much.
But further, those authors use the IAM also below threshold, where there are no data points. There, the choice of parametrization to extrapolate the data does matter. This is https://arxiv.org/pdf/hep-ph/0101037.pdf
The modified IAM was not used there probably because it had not been developed yet, but the authors made a similar approximation subtracting below threshold zeroes and the crossing violations seem to have dropped considerably.

The corresponding paragraph in the text has been extended to reflect these observations.

REPORT:
f) I also have some very minor additional comments/suggestions related to footnotes and references, which I guess are very simple to address:

f.1) In the second note on page 2, there seem to be stronger reasons than that not to adopt Roy.Eqs. As a matter of fact, in the derivation of the IAM, the dispersive integrals also extend to infinity as in Roy equations. Thus, the IAM would be affected by this same caveat. Of course, in the IAM the elastic approximation is taken for the inverse as a simplification. One could think about making the same approximations for Roy eqs. formulated for inverse amplitudes. Thus, whatever happens in the right cut is not the most pressing problem. The real interest in Roy eqs. is to deal with the left cut as precisely as possible and this entails an infinite series over the partial waves in the crossed channels, although in practice reduced to the few first. It is also not clear how to formulate that for the inverses.
RESPONSE: We do not understand the referee very well here. All the IAM needs to get started is the low-energy constants, that only require threshold data. Our understanding from Roy equation analysis is that
the entire physical cut is integrated over, so that data there is necessary. Because this is not reasonably
expected at the LHC (will we even have threshold separations from the SM?) the Roy formulation is not apt
for resonance prediction in a first stage.

REPORT:
f.2) The authors claim that the IAM predicts the rho at 710 MeV with the "pure" ChPT LECS. They should provide a reference for this result and what do they mean by "pure", and why it is more reliable than the other several references about the IAM that the authors quote, where the rho seems to come out fairly well with different values of the LECS in apparently fair agreement with determinations from standard ChPT.

RESPONSE: We have changed the adjective "pure" by the more descriptive "central value from threshold determination". While the referee is right that one can, in hadron physics, describe the rho by shifting the l_i within their band, because the high-energy scale is known, this procedure does not carry to the prediction of an unknown resonance: there one needs to adopt the central value of the l_i uncertainty band.

REPORT:
f.3) Page 1: in the references [5,6] the f0(980) pole is certainly found, but that is not the complete IAM as derived in the manuscript. Maybe the authors may consider adding J.R.Pelaez. Mod.Phys.Lett.A 19 (2004) 2879-2894. Moreover, when discussing the coupled-channel IAM in page 7, and its "less crisp theoretical basis" it would be appropriate to refer the reader to the full IAM works where the theoretical complications were highlighted. I suggest at least: F. Guerrero and J.A. Oller, Nucl.Phys.B 537 (1999) 459-476 as well as A.Gómez Nicola and J.R. Pelaez, Phys.Rev.D 65 (2002) 054009.
f.4) Page one. When mentioning variations of Bethe-Salpeter equations, the authors may consider citing J. Nieves, A. Ruiz Arriola. Nucl.Phys.A 679 (2000) 57-117
f.5) Page 2: when discussing the reconstruction of a heavy scalar particle with the IAM, a relevant reference is: A. Dobado, Phys.Lett.B 237 (1990) 457-462

RESPONSE: We have introduced the references suggested in comments f.3), f.4) and f,5) into the text as requested.

REPORT:
f.6) On page 6: First paragraph of Section 3. Ref. 15 is not about scattering and ref 25 has no dispersion relations. Thus they do not fit well with what is being discussed in the sentence. For the IAM derivation, that function was introduced in refs 13 and 61, in both cases for scattering using dispersion relations.
RESPONSE: We have rearranged the references to include the requested ones and also Truong's historic perspective, and minimally altered the text there to reflect this observation.
We maintain reference [15] because it does partly deal with scattering: it describes various amplitudes, and several treatments to study two-body scattering are discussed. We have now added from it the first reference known to us in which Eq.(8) is introduced, by Harry Lehman, Phys. Lett. 1972, 41, 529.
He did not introduce that equation via the dispersive method as is more common in modern times. The dispersive derivation is advantageous to ascertain the uncertainties but was not the way to discover the equation itself.

REPORT:
Actually, in ref.25 the IAM is introduced as a Padé approximant, an interpretation that the authors may consider mentioning in passing.
RESPONSE: The requested comment has been added to subsec. 4.4, just below Eq.(47) where it probably fits best.

REPORT:
Requested changes
1) Provide an estimate of the highest mass resonances that the IAM could reliably predict, given the systematic uncertainties found here.
2) Provide at least another example of relative uncertainties evaluated for a resonance with different mass than the rho, possibly higher, to illustrate how the relative systematic uncertainty scales.
3) Comment on how the IAM systematic uncertainty compares with the expected uncertainty propagated from the error bars in the low-energy constants if the are obtained from non-resonant physics.
4) Clarify the appearance of M_0 from a chiral Lagrangian and why Eq.81 is a "well behaved {\it chiral} expansion", as well as the convergence of Eq.80, given that f and M_0 can be rather similar in size.
5) Clarify if it is the IAM that fails or is it the low-energy expansion that fails to reproduce the zero at M_0.
6) Clarify or comment the connection between Resonance saturation of the LECS and the presence of a CDD pole. Identify what is the resonance contribution to the LECS in the example.
7) Check if/how the presence of the CDD pole is invalidating the resonance saturation of the LECS in the HEFT example given in appendix A.2.
8) Clarify why/if the cancellation of the real part of the LO+NLO necessarily implies the existence of a CDD pole.
9) Explain why sometimes fπ or sometimes 4π fπ are used as the suppression parameters.
10) Please consider the suggestions for footnotes/references.

RESPONSE: With all these changes we hope that the referee is satisfied and recommends the article for publication.

---

## Editorial Decision

published